# A Spatially Detailed Blue Water Footprint of the United States Economy

*Richard R. Rushforth[1*], Benjamin L. Ruddell[1]*

[1]*School of Informatics, Computing, and Cyber Systems at Northern Arizona University*

*\*Corresponding author*

*Correspondence Emails:*

*Richard.Rushforth@nau.edu*

**Abstract**
This paper quantifies and maps a spatially detailed and economically complete blue water
footprint for the United States, utilizing the National Water Economy Database version 1.1
(NWED). NWED utilizes multiple mesoscale (county-level) federal data resources from the
United States Geological Survey (USGS), the United States Department of Agriculture (USDA),
the U.S. Energy Information Administration (EIA), the U.S. Department of Transportation
(USDOT), the U.S. Department of Energy (USDOE), and the U.S. Bureau of Labor Statistics
(BLS) to quantify water use, economic trade, and commodity flows to construct this water
footprint. Results corroborate previous studies in both the magnitude of the U.S. water footprint
($F$) and in the observed pattern of virtual water flows. Four virtual water accounting scenarios
scenarios were developed with minimum (*Min*), median (*Med*), and maximum (*Max*)
consumptive use scenarios and a withdrawal-based scenario. The median water footprint
($F_{CUMed}$) of the U.S. is 181,966 Mm³ ($F_{Withdrawal}$: 400,844 Mm³; $F_{CUMax}$: 222,144 Mm³; $F_{CUMin}$:
61,117 Mm³) and the median per capita water footprint ($F'_{CUMed}$) of the U.S. is 589 m³ capita$^{-1}$
($F'_{Withdrawal}$: 1298 m³ capita$^{-1}$; $F'_{CUMax}$: 720 m³ capita$^{-1}$; $F'_{CUMin}$: 198 m³ capita$^{-1}$). The U.S. hydro-
economic network is centered on cities and is dominated by use at local and regional scales.
Approximately (58 %) of U.S. water consumption is for the direct and indirect use by cities.
Further, the water footprint of agriculture and livestock is 93 % of the total U.S. water footprint,
and is dominated by irrigated agriculture in the Western U.S. The water footprint of the
industrial, domestic, and power economic sectors is centered on population centers, while the
water footprint of the mining sector is highly dependent on the location of mineral resources.
Owing to uncertainty in consumptive use coefficients alone, the mesoscale blue water footprint
uncertainty ranges from 63 % to over 99 % depending on location. Harmonized region-specific,
economic sector-specific consumption coefficients are necessary to reduce water footprint
uncertainties and to better understand the human economy's water use impact on the
hydrosphere.
**1.  Introduction**

Increasing connectivity through national and global trade has decreased barriers to

economic cooperation while concomitantly increasing the susceptibility of the global economy to
geophysical and meteorological natural hazards (Castle et al., 2014;Diffenbaugh et al.,
2015;Mann and Gleick, 2015;Vörösmarty et al., 2015). Drought – a condition of perceived water
scarcity created by the collision of a dry climate anomaly and excessive human demand for water
that outstrips water availability (Famiglietti and Rodell, 2013; Zetland, 2011) – is one such
natural hazard to which the world is increasingly prone that can impair the production of water-
intensive goods sold in the global marketplace (Vörösmarty et al., 2000;Joseph et al.,
2008;Seager et al., 2007). Without adequate substitutes for water as an input to production, the
economic impact of a drought will propagate beyond local hydrological systems, and dependent
water-intensive industries, into the global economy. Disruptions to the production and
distribution of water-intensive goods, including electricity and other energy sources, have the
potential to spread across seemingly disparate localities over short time periods and are
inherently a coupled natural-human (CNH) system phenomenon (Liu  et al., 2007).
Understanding our vulnerability to these types of events requires a synthesis of network theory,
hydrology, geoscience, and economic theory into a unified food-energy-water (FEW) system
science that is only possible through the novel fusion of comprehensive economic, commodity
flow, hydrologic and geospatial datasets.
Due to global economic connectivity, a drought that diminishes the production and trade
in water-intensive goods has consequences for water resources management worldwide.
Substitutes for drought-affected agricultural products will have to be cultivated elsewhere by
bringing new land under cultivation, intensifying production, or replacing existing crops with
crops no longer viable in the Western U.S. (Mann and Gleick, 2015; Castle et al., 2014; McNutt,
2014). Given the climatic, political, legal, geographical, and infrastructural constraints to
developing new water supplies, which exist to varying extents worldwide, the potential solutions
to systemic global water resources problems now lie in managing the scarcity, equity, and
distribution of existing water resources through the global hydro-economic network rather than
the large-scale development of new, physical sources of water (Gleick, 2003). Further, the
importance of managing the scarcity, equity, and distribution of blue water resources only
increases as rainwater becomes more variable because the majority of water used for food
production in the U.S. is green water (rainwater) (Marston et al., 2018). Physical hydrology and
water supply are mostly localized issues of "blue" physical water stocks and flows of both
human and natural origin. But the global emerges from the local, and actionable information
regarding the scarcity, equity, and distribution of global water resources is attainable only by
mapping the network of hydro-economic connections at a local level, associated with specific
cities, irrigation districts, rivers, and industries. Hydro-economic connections are created through
the trade of water-intensive products and can be measured through virtual water accounting and
water footprinting.
A water footprint is defined as the volume of surface water and groundwater consumed
during the production of a good or service and is also called the virtual water content of a good
or service (Mekonnen and Hoekstra, 2011a). Virtual water, also known as indirect water or
embodied water, has been studied as a strategic resource for two decades as it allows geographic
areas (country, state, province, city) to access more water than is physically available (Allan,
1998; Allan, 2003; Suweis et al., 2011; Dalin et al., 2012; Dang et al., 2015; Zhao et al., 2015;
Marston et al., 2015). Using NWED data, water footprints of production and consumption can be
calculated for U.S. counties, metropolitan areas, and states. A water footprint of production is the
total volume of water consumed with a geographic boundary, including water consumption for
local use less virtual water export (Mekonnen and Hoekstra, 2011b). A water footprint of
consumption is water consumption for local use in addition virtual water import (Mekonnen and
Hoekstra, 2011b).
This paper presents the first spatially-detailed and economically-complete blue water
footprint database of a major country, the U.S., using data from the National Water Economy
Database (NWED), version 1.1. The methodological innovations of NWED lie in trade flow
downscaling through the novel data fusion of multiple U.S. Federal datasets. This process yields
a complete, network-based water footprint database of surface water and groundwater with
flexible geographic aggregation from the county-level to international-level for multiple transit
modes and trade metrics. NWED is economically complete, to the extent possible, since it
utilizes input water data that covers the vast majority of U.S. water withdrawal activities
(Maupin et al., 2014). The service industry is included in NWED although we assume virtual
water flows resulting from the service industries are *de minimus* compared to the commodity-
producing sectors of the economy and thus do not estimate these flows (Rushforth and Ruddell,
2015). NWED contains four consumptive use scenarios – a withdrawal-based scenario, in
addition to minimum, median, and maximum consumptive use scenarios. Currently, NWED is
constrained to blue virtual water flows to focus on potential human-mediated intervention points
in the U.S. hydro-economic network. This article is the publication of record for NWED, which
is currently housed on the Hydroshare data repository (Rushforth and Ruddell, 2017).

With data from NWED, we answer the following research questions:

(1) What is the annual blue water footprint of the United States aggregated by

economic macro-sector and at the spatial mesoscale (county) level?

(2) How does the degree to which a geographic area is urban or rural affect water

footprints, virtual water flows, and net hydro-economic dependencies?

(3) Through which ports does the world access U.S. water resources, and vice versa?

(4) What are the structural and spatial differences between economic sectors' roles in

the U.S. hydro-economy?

(5) What is the current mesoscale uncertainty associated with blue water footprints in

the United States given current data resources?

## 2. Methods

### 2.1. Data

If we are to effectively manage the impacts of drought, and other natural hazards, in the

21st century, we need a detailed quantitative understanding of the world's hydro-economic
network of direct (commodity flow) and indirect connections (virtual water) linking consumers
to producers around the globe. We begin with a blue water footprint that includes saline and
reclaimed water. We include saline and reclaimed water to fully characterize the U.S. hydro-
economy. Specifically, saline and reclaimed water is used as a direct substitute for freshwater use
and is a significant percentage of saline water use for power generation in Florida and the largest
nuclear power plant in the U.S., located in Arizona utilizes reclaimed water. Saline water is also
becoming an important component of municipal water portfolios in California, Texas, Florida
and other states. While the inclusion of saline and reclaimed water in NWED is not a doctrinaire
interpretation of established blue water footprint methodologies, we do believe it is necessary to
include these water types because they are not *de minimus* components of water supply.
Additionally, if there are future constraints to utilizing saline or reclaimed water for power
production, we will be able to anticipate the future added pressure on blue water resources. We
leave green water footprints, and the aquatic ecosystem impacts of water use, to future work.

The hydro-economic network constructed in NWED is built from existing commodity

flow networks and data, specifically the Freight Analysis Framework version 3.5 (FAF)
developed by Oak Ridge National Laboratories for the U.S. Department of Transportation
(Southworth et al., 2010;Hwang et al., 2016), which builds upon the U.S. Commodity Flow
Survey by statistically modelling the flows of several out-of-scope commodity flows, notably
farm-based agricultural flows, natural gas, crude petroleum, and waste. FAF is a detailed U.S.
commodity flow database of 43 commodities traded between 123 freight analysis zones (FAZs),
roughly equivalent to a metropolitan statistical area, over 8 transport modes. The international
component of FAF includes the trade of the 43 commodities by 8 transport modes to 8
international regions. Details of the FAZs, how FAZ-level is derived, commodity classes, and
transport modes have been documented elsewhere and, as such, will not be reproduced in this
paper (Southworth et al., 2010;Hwang et al., 2016;U.S. Bureau of Transportation Statistics,
2017). Note that prior studies have been published using NWED version 1.0 (Rushforth and
Ruddell, 2016). The differences between NWED v 1.0 and 1.1 can be found in the Appendix
(A1).

FAZ trade linkages were disaggregated to component counties/county equivalent areas

using production factors on the production side and attraction factors on the demand side.
Production factors were chosen based on the economic function and product of a sector. For
example, the production factor for agriculture commodities is the area of cultivated irrigated
lands for specific crops (USDA National Agricultural Statistics Service, 2012); the production
factor for the livestock sector is county-level livestock and animal sales for cattle, hogs, and
poultry (USDA National Agricultural Statistics Service, 2012); the production factor for mining
is the number of commodity-specific (e.g., coal, metallic, non-metallic, gravel) mines in a county
(U.S. Geological Service, 2005); and the production factor for the industrial sector is 4-digit
NAICS level employment (Bureau of Labor Statistics, 2012). Currently, NWED uses population
as the only attraction factor (U.S. Census Bureau, 2013), which is as a surrogate for county-level
economic demand for commodities that assumes that all residents consume goods equally.
Population is an adequate attraction factor in the initial NWED version because it is a robust
indicator available for every county in the U.S., but this attraction factor will be subject to further
refinement as new NWED versions are developed.

A harmonization procedure has been developed so that commodities in FAF can be

grouped into larger economic sectors, such as agriculture, livestock, mining, and industrial
sectors to match United States Geological Service (USGS) water withdrawal categories (Maupin
et al., 2014), which NWED utilizes as input water data. Water use categories included in NWED
input data are public supply, domestic, irrigation, thermoelectric power, industrial, mining, and
livestock, which is both livestock operations and aquaculture. Each water withdrawal category is
also further subdivided into groundwater and surface water components as well as freshwater
and saline components. The USGS water data contains water withdrawal data for both the
service and goods/commodity based economy, but NWED currently only contains water
footprint data of the commodity-based economy using a range of empirical, economic sector-
specific consumptive coefficients. Four scenarios are developed from the USGS water input
data: a withdrawal-based scenario (*Withdrawal*) and maximum (*Max*), median (*Med*), and
minimum (*Min*) consumptive use scenarios. Virtual water imports and exports were estimated
using water intensity proxies and detailed in Section 2.10. Future versions will provide detail on
the water-energy nexus, embedded emissions through trade, and the service economy.

Please refer to Appendix (A2) for a Glossary of terms used in this paper and to describe

aspects of the NWED method and analysis in full detail.
**2.2.      Temporal Representativeness**

Both FAF data and USGS water withdrawal data are collected every five years. However,

FAF data is published for years ending with 2 and 7 (i.e., 2002, 2007, and 2012) and USGS data
is published every half decade (i.e., 2005, 2010). NREL ReEDS modeled power flow data is
available biennially from 2010 to 2050 (Eurek et al., 2016). The current version of NWED
utilizes FAF data published for 2012 and USGS water withdrawal data published for 2010.
Water withdrawal data for 2010 captures the beginning of Texas-North Mexico drought that
lasted from 2010 to 2011 (Seager et al., 2014) and is situated between significant droughts in
California between 2007 and 2009 (Christian-Smith et al., 2015) and 2011 to 2014 (Seager et al.,
2015). It is possible that these two hydrologic droughts increased water groundwater withdrawals
and consumption in the U.S. during 2010 calendar year in the southwestern and southcentral U.S.
These data were used as the basis of the county-level U.S. National Water Economy Database
version 1.1 (NWED). The results of this NWED data product are limited in representativeness to
roughly the 2010 – 2012 post-recession timeframe but are not precisely linked to a single year.

The current version of NWED has an annual resolution due to a lack of comprehensive,

sub-annual county-level data. While economic data are available at sub-annual timescales, often
quarterly, water withdrawal data are not. However, annual water withdrawal and consumption
data could be disaggregated to the month scale using median monthly demand curves (Archfield
et al., 2009; Weiskel et al., 2010). This lack of data availability does present challenges because
there are substantial sub-annual fluctuations in water withdrawal and consumption. Water
demands for agriculture and power are highly seasonal and neither the beginning nor the end of a
drought coincides with calendar years. For example, the Texas-North Mexico drought began in
the latter half of 2010 (Seager et al., 2014). As we further develop NWED, we will develop
methods to address this shortcoming, but for now are limited to the annual timescale.

**2.3.    Geography of NWED**

The county-scale of geography and annual-scale of time are the appropriate scales of

aggregation for a nationally-scoped water footprint analysis in the U.S. given the available water
withdrawal and commodity flow data. For the purposes of planning, policy, and law, especially
in the absence of larger cities, counties and county equivalents are socio-political units that
effectively define the "local" scale of U.S. society and the economy. Additionally, most services
are consumed locally within the county where they are produced. In rural areas, a county is an
aggregation of socio-economically similar small towns and agricultural areas. In urban areas, a
county is more socio-economically diverse, but its statistical data are dominated by a single
major metropolitan area and the county is, therefore, representative of that metropolitan area.
While the largest metropolitan areas in the U.S. cover several counties and range from a half
million people to over 10 million, counties can still capture the economic diversity within the
metropolitan area.
The FAF FAZ is a group of counties that roughly comprise a metropolitan area, reflecting
the fact that the commodity distribution infrastructure of the United States is organized as a
spoke-and-hub network with major metropolitan areas and their distribution centers as hubs, thus
necessitating the need to develop a disaggregation method. FAZ were disaggregated to the
county level using best practices from the literature: population as an attraction factor on the
demand side and employment levels, the number of agricultural and livestock operations, and the
number of commodity-specific mining facilities on the production side (Viswanathan et al.,
2008;Bujanda et al., 2014;Harris et al., 2012;De Jong et al., 2004). These data allow for the
development of a robust set of disaggregation factors that ensure the production of a commodity
occurs only where it is physically and economically possible.
Standardized water use data and water stress data are available nationwide at the county-
scale but do not typically exist at finer scales. A spatial unit coarser than the county will fail to
capture the dominant hydrological and socio-economic patterns in the water footprint, and a finer
spatial unit of analysis is not yet possible due to a fundamental lack of consistent, national data at
those scales. If finer scale or more up-to-date data do exist, those data may not be consistent with
national data, so consistency becomes a primary quality control issue (Mubako et al., 2013).
Nonetheless, sub-annual and sub-county scale water use, economic production, water stress, and
trade data are all needed to achieve a higher level of detail in the water footprint.

**2.4.    NWED Naming Convention**
The general form of a trade linkage ($T$) in the FAF database is a commodity ($c$) that flows
from an origin FAZ ($O_o$) to a destination FAZ ($D_d$) over a domestic transport mode ($k_{dom}$)
represented as tons ($t$), currency ($\$$), and ton-miles ($tm$), where $o$ and $d$ are indices for the 123
FAZ. Additionally, each *c* is associated with a broader economic sector (*s*) that corresponds to
the USGS water withdrawal categories. International imports and exports originate from and
terminate at one of 8 international origin ($O_I$) and destination ($D_E$) zones via an international
transport mode ($k_{int}$). For an import, a *c* is produced in an international region ($O_I$) and flows
through a port of entry ($O_o$) and then to a $D_d$ of final consumption. For an export, a *c* is produced
in a $O_o$ and then exits the U.S. through a port of exit ($D_d$) for consumption in an international
region ($D_E$). Domestic, import and export trades can be also classified by a trade type index (*f*)
Therefore, a trade linkage of a commodity in terms of *t, $,* and *tm* between an origin zone and
destination, which may not include a foreign region, can be represented as
$T_{O_I,O_o,D_d,D_E,k_{int},k_{dom},c,f}(t,\$,tm)$. NWED builds upon FAF by further disaggregating $O_o$ and $D_d$ to
origin ($I_n$) and destination counties ($J_m$), respectively, and by adding virtual water, represented
generally as (*VW*). Each row in NWED is trade linkage, $T_{O_I,O_o,I_n,J_m,D_d,D_E,k_{int},k_{dom},c,f}$, with a
corresponding flow of *t*, *$*, *tm*, and *VW* that can be aggregated by any combinations of index
$O_I \rightarrow f$. However, we drop all of these subscripts for a simpler derivation of the NWED
disaggregation algorithm. NWED retains data for transport mode, tons, currency, and ton-miles
as there are NWED use cases outside of virtual water accounting that may utilize mode-specific
data or data on *$* or *tm* flows.

**2.5.    Water Footprint of a Geographic Area**

248        The water footprint of a geographic area ($F_{Total}$) is the sum of the direct water use (*WU*),

virtual water inflows ($VW_{In}$), and virtual water outflows ($VW_{Out}$) (Hoekstra et al., 2012). For
example, in NWED, the water footprint of withdrawals of geographic area for all economic
sectors is $F_w = WU_w + VW_{In,W} - VW_{Out,W}$ or alternatively $F_{Total} = WU_W + VW_{Net,W}$,
where $VW_{Net,W} = VW_{In,W} - VW_{Out,W}$. The per-capita footprint is $F`$ and is calculated by
dividing $F$ by the population of the county. Within NWED, the sum of $F$ across all domestic
trade in the U.S. yields $VW_{In,W} = VW_{Out,W}$ to ensure the water balance is conserved. $F$ and each
of its components are reported for each economic sector within each county in the U.S. in
NWED. The derivation of $VW_{In,W}$ and $VW_{Out,W}$ are shown in section 2.6 – 2.8.

**2.6.    Disaggregating Domestic Trade Flows to the County-Level**

259          The disaggregation method proceeds from the origin side ($O$), disaggregating to origin

counties ($I$), and then to the destination side ($D$), disaggregating to destination counties ($J$). Each
$O$ contains a distinct set of one or multiple origin counties ($I_n$), where $I_n \in O$, and each $D$
contains a distinct set of multiple destination counties ($J_m$), where $J_m \in D$. Further, each county
($n$ or $m$) within each $O$ and $D$ has a unique production factor ($PF$) and attraction factor ($AF$) for
each economic sector and, where supported by data, each commodity produced in that county.
Each $I$ and $J$ can be defined as distinct set of unitless $PF$ or $AF$ factors for each commodity,
$\{I_n: PF_{c1}, PF_{c2}, \ldots, PF_{c43}\}$ and $\{J_m: AF_{c1}, AF_{c2}, \ldots, AF_{c43}\}$, repectively. Therefore, any $O_o$ or $D_d$
can be represented by a column vector of $PF_c$ or $AF_c$ corresponding to the $I_n$ or $J_m$ that belong to
$O_o$ or $D_d$. Given that the $PF_c$ or $AF_c$ define the proportion of production capacity and demand
attraction a county has within a $O_o$ or $D_d$, the sum of the $PF_c$ or $AF_c$ for a given $O_o$ or $D_d$ must be
equal to 1 to conserve mass. Therefore, for a given commodity ($c$) with an associated sector ($s$)
and $t$, $\$$, and $tm$ over 8 transport modes, $k$,

272          (1) $O_{o,c} = \begin{bmatrix} I_{1PF_c,O_{o,c}} \\ I_{2PF_c,O_{o,c}} \\ \vdots \\ I_{nPF_c,O_{o,c}} \end{bmatrix}$ or $D_{d,c} = \begin{bmatrix} J_{1AF_c,D_{d,c}} \\ J_{2AF_c,D_{d,c}} \\ \vdots \\ J_{nAF_c,D_{d,c}} \end{bmatrix}$, where $\sum_n O_o = 1$ and $\sum_m D_d = 1$.

Disaggregating production from a $O_o$ that contains counties $I_{1\rightarrow n}$, $O = \{I_1, I_2, ..., I_n\}$ for a
$c$ proceeds as follows:

$$
(2)\ T_{O_o,D_d,c} \times
\begin{bmatrix}
I_{1PF_c,O_o,c} \\
I_{2PF_c,O_o,c} \\
\vdots \\
I_{nPF_c,O_o,c}
\end{bmatrix}
=
\begin{bmatrix}
T_{I_1,D_d,c} \\
T_{I_2,D_d,c} \\
\vdots \\
T_{I_n,D_d,c}
\end{bmatrix}
$$


Solving Equation 2 over all $O_o$ for each commodity disaggregates FAZ-level commodity
production to the county-level – from 123 origin FAZs ($O_o$) to 3,142 origin counties ($I_n$). A
quality control is performed to ensure that no additional mass, currency, or ton-miles are
produced for all commodities across all $O_o$. After the production-side disaggregation, 3,142
origin counties are linked with 123 FAZ destinations via trade of commodities ($c$).
Similarly, the goal of the demands-side disaggregation is to disaggregate flows to 123
FAZ to 3,142 counties; however, instead of the relative abundance of industries that produce a
specific commodity to disaggregate production, population is used as a simple measure of a
county's attraction (demand) of a commodity within a FAZ. It follows that disaggregation on
demand side of the O-D trade linkage follows a similar process.
For a $D_d$ that contains counties $J_1$ to $J_n$, $D_d = \{J_1, J_2, ..., j_n\}$ for $g$ produced in an origin
county, $I_n$, disaggregation proceeds as follows:

$$
(3)\ T_{I_n,D_d,c} \times
\begin{bmatrix}
J_{1AF_c,D_d} \\
J_{2AF_c,D_d} \\
\vdots \\
J_{nAF_c,D_d}
\end{bmatrix}
=
\begin{bmatrix}
T_{I_n,J_1,c} \\
T_{I_n,J_2,c} \\
\vdots \\
T_{I_n,J_m,c}
\end{bmatrix}
$$


At this point, quality control is performed to ensure that no new mass, currency, or ton-
miles are erroneously introduced for all commodities across all $O_o$ and $D_d$. Performing this
disaggregation step across all $I_n$ disaggregates the flows of $c$ in terms of $t$, $\$$, and $tm$ to be
between 3,142 origin counties and 3,142 destinations counties over 8 potential transport modes,
$k$.

International flow disaggregation follows the same process; however, the 8 world regions

are not disaggregated further and import flows are not further disaggregated into surface water
and groundwater. After, import and export flows are disaggregated each world region is
connected via a production of consumption trade flow with one of 3,142 U.S. counties flowing
through a port of entry or exit.

**2.7.    Assigning Virtual Water Flows to Trade Flows**

Economic sectors ($s$) in the FAF database were aligned with water withdrawal sectors

($WU_s$) using the detailed Standardized Classification of Transported Goods (SCTG) definitions
of commodity groups (US Census Buearu, 2006; Dang et al., 2015). County-specific, sector-
level water intensities ($WI_{I_n,S,W_{Total}}$) were calculated as the quotient of county-specific, sector-
level water withdrawals ($WU_{I_n,S,W_{Total}}$) and county-level, sector-specific commodity production
($\sum_{D_d,c} T_{I_n,D_d,c}$) and have the units $Mm^3$ $t^{-1}$. In the initial step of calculating $WI_{I_n,S,W_{Total}}$,
groundwater and surface water withdrawals are summed to a total sector-level water withdrawal
figure for each county ($WI_{I_n,S,W_{Total}}$). Virtual water flows are disaggregated back to groundwater
and surface water fractions in a later step.

(4)     $WI_{I_n,S,W_{Total}} = WU_{I_n,S,W_{Total}}/\sum_{D_d,c} T_{I_n,D_d,c}$

The resulting $WI_{I_n,S,W_{Total}}$ can be interpreted as the average withdrawal-based water

intensity of sector-level production.

Next, $WI_{I_n,S,W_{Total}}$ were multiplied by the corresponding $T_{I_n,J_m,c}$ to arrive at the virtual

water flows by county and commodity by transport mode.
(5)      $VW_{I_n,J_m,c,W_{Total}} = WI_{I_n,s,W_{Total}} \times T_{I_n,J_m,c}$
The $VW_{I_n,J_m,c}$ that results from this process assigns water withdrawals to a commodity
based on the tons of a $c$ within a county according to the disaggregated FAF data. Future
versions of NWED will refine this process with additional commodity specific water intensities,
as explained further in section 2.4.
For notational clarity, when $VW_{I_n,J_m,c,W_{Total}}$ is summed for all unique origin counties ($I_n$)
the term is simplified to $VW_{Out,Total}$. Conversely, when summed for all unique destination
counties ($J_m$) the term is simplified to $VW_{In,Total}$. Additionally, $WU_{I_n,s,Total}$ summed over all
sectors for all unique counties becomes $WU_{W_{Total}}$. This notation also holds true for
consumption-based virtual water flows.
Minimum (*Min*), median (*Med*), and high (*Max*) water consumption scenarios for each
sector in each county were determined by multiplying $WU_{I_n,s,W}$ by the corresponding sector-
level minimum, median, and high consumption coefficients developed by the USGS (Shaffer and
Runkle, 2007). Only the methodology for *Med* consumption scenario is shown below since both
the *Min* and *Max* consumption scenarios follow an identical calculation process.
(6)      $WI_{I_n,s,CU_{Med,Total}} = (WU_{I_n,s,W_{Total}} \times CU_{Med,S})/\sum_{D_d,c} T_{I_n,D_d,c}$
(7)      $VW_{I_n,J_m,c,CU_{Med,Total}} = WI_{I_n,s,CU_{Med,Total}} \times T_{I_n,J_m,c}$
Owing to these consumption coefficients being developed for the Great Lakes Region, and
climatically similar states, the consumption-based virtual water flows in NWED are preliminary
and serve as placeholders until region- or county-specific and sector-level consumption
coefficients have been developed for the U.S.
Since the USGS water withdrawal data contains data on groundwater and surface water
withdrawals for each sector within each county, $VW_{I_n,J_m,c,CU_{Max,Total}}$, $VW_{I_n,J_m,c,CU_{Med,Total}}$, and
$VW_{I_n,J_m,c,CU_{Min,Total}}$ are split into groundwater and surface water components be multiplying each
by the county-specific, sector-specific groundwater withdrawal percentage ($GW_{I_n,s,pct}$) and
surface water percentage ($SW_{I_n,s,pct}$). The process is shown below for $VW_{I_n,J_m,c,s,t,k,CU_{Max}}$.

(8)    $VW_{I_n,J_m,c,CU_{Max,SW}} = VW_{I_n,J_m,c,CU_{Max,Total}} \times SW_{I_n,s,pct}$

(9)    $VW_{I_n,J_m,c,CU_{Max,GW}} = VW_{I_n,J_m,c,CU_{Max,Total}} \times GW_{I_n,s,pct}$

After this step, there is a final mass balance check to ensure NWED freight totals match

underlying FAF data and water data match underlying USGS data. NWED contains data
detailing 3,142 counties trading 43 commodities with 3,142 counties, as well as 8 world regions,
over 8 transport modes and each commodity trade linkage is measured by 15 metrics (The full
list of metrics is in the Appendix, A3).

**2.8.    Power Flow Estimation and Disaggregation**

The flow of the electricity commodity is not like other commodity flows. There is no

mass moved from point A to point B, and there is not a contract associated with such a flow. The
concept of power flow is as philosophical as it is physical. However, we know some of the
geometrical properties of the power grid. The grid is comprised of the U.S., at the first level of
aggregation, of three interconnections: the Western Electricity Coordinating Council (WECC),
the Eastern Interconnection (Eastern), and the Electric Reliability Council of Texas (ERCOT),
with little transmission of electricity between them. Interconnections do not obey county or state
boundaries, or even national borders; Mexico and Canada are participants in WECC and Canada
in the Eastern. At the second level of aggregation, the grid is comprised of 134 balancing
authorities within which a single authority has responsibility for maintaining a balance between
supply and demand and managing power quality. Balancing authorities trade power between
themselves, but strongly manage these transmission corridors. Within a balancing authority,
there is a mixture of power generators, transmitters, and distributors that participate in a
complicated web of heretofore uncatalogued contracts using a complex interconnected machine
that maintains a constant voltage potential and frequency under variable loads. Adding to this
complication is the absence of standardized mesoscale, coupled power generation, transmissions,
and power consumption datasets.
Given this unusual situation, we know of at least three methods for estimating the
destination and routing of electricity. First, because we can assume there is little trade across an
interconnection's boundary, a "mass balance" could be applied within an interconnection's
subregions, allocating consumption first to the local generator's region and then in proportion to
estimated demand in other regions (e.g. Ruddell et al., 2014). This method is not physically
realistic because it ignores transmission constraints and balancing regions but may be a useful
approximation especially at coarser spatio-temporal scales. A second method is to follow
contracts and payments for electricity and power services. This method provides the closest
analogy to the commodity flow model, but the contract and payment data is not currently
available. A third method is to perform power flow modeling on a spatio-temporally precise
node-network model of the grid that incorporates detailed information about generators, demand
patterns, and their economics to simulate power flows as an analogy to commodity trade. We use
balancing region power flow modeling for NWED 1.1, disaggregated to the county scale using
population.
The power flow data used in NWED is an existing published dataset produced using the
Regional Energy Deployment System (ReEDS), which is a long-term power flow model to
evaluate capacity-expansion, technology deployment, and infrastructure deployment in the
contiguous U.S (Macknick et al., 2015;Eurek et al., 2016;Cohen et al., 2014). Only for the
electrical power production sector, NREL data on water withdrawal and consumption data were
used instead of USGS water withdrawal data to estimate the water withdrawal and consumption
associated with power generation and flow (Macknick et al., 2012; Macknick et al., 2015).

ReEDS data contains both power generation by balancing authority and power inflows

and outflow between balancing areas over sub-annual time periods. Balancing authorities are
areas larger than counties. To harmonize with NWED and disaggregate ReEDS data from the
balancing authority to the county-level, the model's production numbers are disaggregated
proportionally using the heat content of fuel consumption for electricity for each county's power
plants (Energy Information Administration, 2017) and electricity demand is disaggregated
proportionally by population.

In addition to error introduced in disaggregation, power wheeling within balancing

regions is a significant portion of power flow, and this is another source of error (Bialek,
1996a;Bialek, 1996b;Bialek and Kattuman, 2004). To help compensate for the effect of wheeling
on the water footprint of electricity, the water intensity of a power outflows from each balancing
area was taken as the source-weighted average of the water intensity of power generation and
power inflows. Therefore, virtual water outflows from a county in NWED 1.1 is the virtual water
outflow associated with wheeled power through a balancing area (including power originating
from this area's generation) in addition to virtual water outflows associated with power
generation within that county. Taking into account these modifications to the standard virtual
water methods employed elsewhere, virtual water flows were estimated according to the methods
in sections 2.5 – 2.6.

**2.9.    Urban-Rural Classification**
Each county in the U.S. can be categorized using numerous classification schemes. For this
paper, and for the purpose of understanding rural-to-urban transfers of virtual water in the U.S.,
we have classified each county in NWED by the National Center for Health Center for Health
Statistics (NCHS) Urban-Rural Classification Scheme for Counties (Ingram and Franco, 2012).
Within this classification scheme, counties are first separated into metropolitan and non-
metropolitan counties. Metropolitan, or urban, counties are then further classified as Large
Central Metro counties (*Central*), Large Fringe Metro counties (*Fringe*), Medium Metro counties
(*Medium*); and Small Metro counties (*Small*). Generally, large counties have greater than 1
million people; medium counties have between 250,000–999,999 people; and small counties
contain less than 250,000 people. Non-metropolitan, or rural, counties are divided into
Micropolitan (*Micro*) counties (population between 10,000–49,999 people) and non-core
counties are counties with a population too small to be considered micropolitan counties. Each
county-to-county trade linkage has been classified and aggregated by the NCHS Urban-Rural
Classification Scheme for Counties to understand urban to rural virtual water transfers (Section

3.1).


**2.10.    Simplifying Assumptions and Limitations**
NWED water footprints, by necessity, are multiple water sources and types beyond
simply groundwater and surface water. Saline and brackish water are non-trivial components of
U.S. water use, comprising about 14% of total water withdrawals – specifically, power
generation in Florida, mining in Texas and Oklahoma (Maupin et al., 2014). Thus, saline water is
a non-trivial component of the U.S. hydro-economy. For example, only 71 % of power
generation in the U.S. is from freshwater sources and the remaining fraction of water use for
power generation is comprised of saline, brackish, and reclaimed water (Maupin et al., 2014).
Neglecting non-freshwater sources would underestimate the water intensity of the power grid.
Reclaimed water is a direct substitute for fresh water, and brackish water is a substitute in some
cases, so it is difficult to draw a clear line between included and excluded water withdrawals.
Considering the entire U.S. hydro-economy, 15 % of water withdrawals are saline. However, the
inclusion of non-freshwater sources does not impact the agricultural virtual water flows as no
saline water withdrawals are reported in this sector. For simplicity in this paper, commodity-
based virtual water flows are reported as 'blue water' even though we incorporate additional
types of water beyond freshwater. Power flow-based virtual water flows are presented summed
over all water types - not just freshwater. The freshwater footprint of electricity is somewhat
smaller than the total water footprint, and this difference is larger on the coasts and in the West.

The current version of NWED uses national average U.S. water use efficiencies to

estimate international virtual water flows. The first reason for this choice is data consistency.
While the USGS water use data does contain some interstate variability due to data reporting
methods, the variability is no doubt far smaller than international variability in data reporting
methods among countries that mostly lack formal water census programs. Secondly, the U.S. is a
large, and geographically, agronomically, climatically, and economically diverse country; water
use efficiencies vary dramatically from region-to-region and sector-to-sector. This internal
variability captures a large range of the world's variability. Third, the U.S.'s water use efficiency
is near the middle of the international range. According to World Bank data, the U.S.'s average
per GDP water use productivity between 2005–2015 was in the 65th percentile of reporting
countries (World Bank, 2017). Fourth, the USGS presents comprehensive water withdrawal data
for all types of mining products, which are an important import to the U.S. Finally, since NWED
is U.S.-centric, this method normalizes virtual water flows to U.S. water efficiencies, allowing
for a 1:1 equivalency between the volume of virtual water traded by the U.S. to the volume of
virtual water flowing internally (Rushforth et al., 2013). In other words, 1 unit of water use
outsourced from the U.S. via virtual water imports directly offsets and substitutes for 1 unit of
water used in the consuming U.S. location; this is a useful comparison also employed by other
studies in the literature (Mayer et al., 2016).

From the USGS water withdrawal data, we use total, surface water, and groundwater

withdrawals from each county. The sum of all withdrawals in a county is the direct use
component of that county's Water Footprint ($\sum_s WU_{I_n,s,W_{Total}}$, $or$ $WU_{Total}$). $WU_{Total}$ is the sum
of agriculture ($WU_{I_n,Ag,W_{Total}}$), not including the irrigation of golf courses; industrial
($WU_{I_n,Ind,W_{Total}}$), which is estimated by taking the sum of industrial withdrawals and the
difference between water withdrawal for public supplies and domestic uses by water systems;
mining ($WU_{I_n,Min,W_{Total}}$); and livestock, which includes livestock and aquaculture withdrawals
($WU_{I_n,Liv,W_{Total}}$). $WU_{I_n,W_{Total}}$ is also known as the Water Metabolism of a county (Kennedy et
al., 2015). Total, surface water, and groundwater water footprints within a county match the
standard Water Footprint Accounting definition of the water footprint of a geographic area
(Hoekstra et al., 2012). For withdrawal-based water footprints, we assume 100 % consumptive
use (consumption coefficient $CU = 1$), forcing USGS-estimated water withdrawals equal to the
direct use component of the Water Footprint, $WU$. Sector-level consumption coefficient data do
exist, but these data are specific to the Great Lakes region of the U.S., and climatically similar
states, and have large uncertainty ranges (Shaffer and Runkle, 2007). Due to the large
uncertainties involved with the consumption coefficients, we have attempted to estimate the
uncertainty associated with consumption by using three consumption coefficients for each sector
– a minimum (*Min*), median (*Med*), and maximum (*Max*) (Table 1). The uncertainty introduced
by the consumption coefficients, and how it propagates when applied over a trade network, is
presented in Section 3.5. Future work can augment NWED by developing more accurate
consumption coefficients estimate for all counties, or regions, in the U.S. for all economic
sectors. NWED contains the following assumptions regarding water use categories: (1) USGS
aquaculture and livestock are combined into one category since specific commodity codes
includes both live meat and fish and because aquaculture is a *de minimus* water use compared to
livestock; (2) USGS industrial water supply is calculated to include the component of public
water supply that is not for domestic household consumption in addition to industrial water
withdrawals; (3) each water use category includes both publically-supplied and self-supplied
withdrawal figures; and (4) while virtual water flows associated with water use categories
outside the scope of the FAF commodity flow database are neglected, direct water use is
accounted.

With respect to (4), this specifically includes flows of services and labor across county or

regional lines (Rushforth and Ruddell, 2015). There is a substantial absolute error introduced by
zeroing virtual water flows out from counties that export services and FAF-ignored goods, and
this error causes urban areas' net water footprints to be overestimated (and rural areas' to be
underestimated by exactly the same amount). Water balances *WU* are unchanged. However, this
error is small in relative terms because these sectors are a small part of total virtual water flows
when compared with agriculture, power, and major industry. Labor and services are consumed
largely within their county of production. Important exceptions may possibly include the
financial services sector, which tends to be national and global in its trading patterns.
A limitation in the underlying FAF data is that an assumption must be made that
commodity production occurs at the origin and commodity consumption occurs at the
destination. Therefore, we must assume that there are no pass-through commodity flows. To the
extent possible in the underlying data, this is controlled for at international ports because pass-
through commodity flows are identifiable from commodity flow to or from the city in which the
port is located. However, domestic pass-through commodity flows are not identified in the
current version of NWED. A method to estimate pass-through commodity flows using input-
output methods is under development and will be included in the next version of NWED.
Future iterations of the NWED power flow dataset will utilize purpose-built node-
network power flow models developed at the county-level to differentiate between power
outflows into generated power and wheeled power for each county.
**3.   Results**
**3.1.   U.S. Water Footprint Statistics**
The median annual water footprint, $F_{CUMed}$, of the U.S. is 181,966 Mm³ ($F_{Withdrawal}$:
400,844 Mm³; $F_{CUMax}$: 222,144 Mm³; $F_{CUMin}$: 61,117 Mm³). On per-capita basis, the median U.S.
water footprint ($F'_{CUMed}$) is 589 m³ capita$^{-1}$ ($F'_{Withdrawal}$: 1298 m³ capita$^{-1}$; $F'_{CUMax}$: 720 m³ capita$^{-1}$;
$F'_{CUMin}$: 198 m³ capita$^{-1}$). Counties with the largest $F_{CUMed}$ are often metropolitan areas with large
populations or regionally-significant cities with neighboring counties that are heavily agricultural
– Los Angeles County, California (L.A.); Harris County, Texas (Houston); Ada County, Idaho
(Boise); Maricopa County, Arizona (Phoenix); and Fresno County, California (Fresno) (Fig. 1;
withdrawal-based results are presented in the Supplemental Information.). On a per capita basis,
the U.S. water footprint is smallest for urban areas, where $F'_{CUMed, Urban}$ is 282 m³ capita$^{-1}$
($F'_{Withdrawal,Urban}$: 828 m³ capita$^{-1}$; $F'_{CUMax,Urban}$: 399 m³ capita$^{-1}$; $F'_{CUMin,Urban}$: 97 m³ capita$^{-1}$) and
largest for rural, agricultural counties $F'_{CUMed, Agriculture}$ is 1,053 m³ capita$^{-1}$ ($F'_{Withdrawal-Basis,}$
$_{Agriculture}$: 1,927 m³ capita$^{-1}$; $F'_{CUMax, Agriculture}$: 1,217 m³ capita$^{-1}$; $F'_{CUMin, Agriculture}$: 344 m³ capita$^{-1}$).

NWED results are comparable to previous water footprint studies for the U.S. For

example, Mekonnen and Hoekstra estimated the U.S. blue and grey water footprint to be 320,496
Mm$^3$ and 874 m$^3$ capita$^{-1}$ (Mekonnen and Hoekstra, 2011a), which is the closest equivalent to the
water sources used NWED. The Mekonnen and Hoekstra U.S. water footprint figures sit roughly
between the $CU_{Max}$ and withdrawal-based (CU = 1) NWED scenarios. Further, results from
NWED corroborate previous studies in both the magnitude of the U.S. water footprint and in the
observed pattern of virtual water flows to cities concentrated in water-intensive irrigated
agricultural and industrial goods (Rushforth and Ruddell, 2015; Zhao et al., 2015; Hoekstra and
Wiedmann, 2014). Vital water footprint statistics are presented in Table 2 for the U.S. in addition
to urban (*Central, Fringe, Medium*) and rural (*Small, Micro, Non-Core)* counties.

Counties in California's Central Valley – Fresno County and Tulare County located in

the southern part of the Central Valley – have the largest virtual water outflows of any county in
the U.S. Overall, the western U.S., the High Plains, the Mississippi Embayment, Texas Gulf
Coast, and Florida provide the U.S. with virtual water exports. Coincidentally, all these source
regions are highly prone to either drought or flooding (production-level uncertainty). Large
virtual water outflows are often counterbalanced by nearby virtual water inflows within the same
county (Fresno County, California) or region, as is the case with Fresno County, California, Pinal
County, Arizona (net outflows from irrigated agriculture) and neighboring Maricopa County (net
inflows to the Phoenix Metropolitan Area) and Brazoria County, Texas (net outflows from
irrigated agriculture) and Harris County (net inflows to the Houston Metropolitan Area) in
Texas. In general, we find that the water supply chain, especially the step of the chain bringing
agricultural products from the farm to handling and processing facilities where these products
become 'food' is mostly local and regional with a smaller but still significant transnational and
international water supply chain.

**3.2 Urban Dependencies on Rural Virtual Water**
Circular virtual water flows – virtual water flows that originate and terminate within the
same county – are highest for urban counties (Fig. 2). Conversely, rural counties often have
small water footprints regardless of the presence of a large water-intensive industry, because
rural populations do not consume the majority of the goods produced in those regions. If such an
industry were present in a rural county, much of the water withdrawn flows out of the county as
virtual water, thus counterbalancing the large withdrawals. Counties that are in the middle of the
urban-rural spectrum, often a medium-to-small metropolitan area, rely heavily on agricultural
products as an economic input and tend to have the largest virtual water inflows of all U.S.
counties. Medium to small cities tend to be food processing hubs where farm goods are
transformed into 'food.' and NWED assigns irrigated agricultural blue water footprints to these
hubs. We recognize that this framing of the economy emphasizes different parts of the supply
chain than previous studies and are developing methods for supply chain harmonization.
The central counties of large metropolitan areas (*Central)* tend to source virtual water
equally across the urban-rural spectrum with a slight increase in virtual water sourcing from
more medium metropolitan areas and rural counties. However, there is a comparatively small
return flow of virtual water from large metropolitan areas back to counties with smaller
populations (Table 3). Instead, virtual water originating from counties associated with large
metropolitan areas tend to remain within that county as a circular flow or flow to other large
metropolitan areas, enlarging the net VW inflow of large metropolitan areas.

One such county is Maricopa County, the central county of the Phoenix metropolitan

area, which  a "local water" hotspot where most of the water used in the community "stays local"
in the form of locally consumed virtual water flowing to other users in the same community.
This means the community is employing its blue water resources primarily for the hydro-
economic benefit of its local consumers and businesses. It also means that this community's
dependency on its own local water resources is amplified through self-dependence, so any
disruption to local water supplies in Phoenix will have a positive feedback loop on that city's
economy (Rushforth and Ruddell, 2015). The Phoenix metropolitan area is notable as a major
city and population center that is simultaneously a large user of irrigation water for the
production of agricultural commodities, including locally consumed food products. Phoenix is
also relatively isolated geographically from other metropolitan areas and therefore keeps more of
its metropolitan area's virtual water within the local boundary, unlike east coast cities where
intra-metro trade and virtual water flows are more prevalent.

Counties that are associated with medium-sized metropolitan areas (*Medium)* break from

large cities' and their fringes and take on a different role in the system. While medium
metropolitan areas are by no means small, with a population between 250,000–999,999, they are
often co-located with large agricultural areas. For example, Ada County, Idaho (Boise metro
area), Fresno County, California (Fresno metro area), or Kern County, California (Bakersfield
metro) are all counties that contain medium-size metropolitan areas that are co-located with
intense agricultural production. In these counties, virtual water tends to be sourced from counties
that are as rural as the place of consumption or more rural. Medium-sized metropolitan areas, in
particular, are the largest destination of virtual water from rural America while also being one of
the largest sources of virtual water for the U.S., especially large metropolitan area – effectively
linking rural and urban counties. The medium-medium urban connection is the largest link in the
U.S. virtual water flow network, and this link is dominated by the heavy industrial and bulk
agricultural and processed food goods that do not tend to be produced by highly rural or densely
urban areas. On a per capita basis, the Medium class of city is the core of the U.S. hydro-
economic network. County-level virtual water flow data show that there is an urban-rural divide,
suggesting that there is a fundamental difference in the roles of large urban areas, medium urban
areas, and more rural communities in the U.S. hydro-economic network.
In the U.S. hydro-economy, economic sectors have different structural roles as either a
virtual water sink or source depending on the degree to which a county is rural or urban.
Structurally, the agricultural sector is the bulk of the rural-to-urban transfer of virtual water
(59,119 $Mm^3$), but rural-to-rural and urban-to-urban virtual water flows are also significant
(53,731 $Mm^3$ and 27,743 $Mm^3$, respectively). While similar, the livestock sector constitutes a
minority of the rural-to-urban transfer of virtual water (6,100 $Mm^3$) but has little to no impact on
virtual water exports. Due to the structure of the underlying commodity flow dataset, the
livestock sector only includes on-site water consumption at livestock operations. Inclusion of
water usage for livestock feed would, no doubt, increase virtual water transfers related to the
livestock sector and a method to do so is under development for the next NWED version. The
mining sector is more geographically-dependent and regional on the location of resources and
infrastructure. Therefore, while rural-to-urban virtual water flows are the largest within this
sector (337 $Mm^3$), rural-to-rural and urban-to-urban virtual water flows are also prominent (175
$Mm^3$ and 165 $Mm^3$, respectively). In the power sector, the largest virtual water flow is from
rural-to-rural (159 $Mm^3$) followed by urban-to-urban (22 $Mm^3$) and rural-to-urban (13 $Mm^3$).
While there are large water withdrawals associated with the power sector, water consumption is
relatively low compared to other sectors. Since the results presented are for the $CU_{Med}$ scenario,
the power sector virtual water flows are small relative to the other sectors. Finally, the industrial
sector is primarily urban-to-urban virtual water transfers. Rural-to-urban virtual water transfers
would only become more pronounced if *Medium* metropolitan areas were considered to be rural
counties. While there is subjectivity to whether a county is rural or urban, especially in the
middle of the urban-rural spectrum, the predominant flow of virtual water is from rural counties
to urban counties.

**3.3 U.S. International Virtual Water Imports and Exports**
Overall, the U.S. is a net virtual water exporter, which qualitatively agrees with the
findings from previous international virtual water flow studies (Water Footprint Network, 2013);
the virtual water balance of the United States is -4,693 $Mm^3$. However, while our virtual water
balance results agree qualitatively with previous studies, the magnitude of virtual import and
export in NWED is an order of magnitude lower than previously published international virtual
water trade data (Water Footprint Network, 2013). Potential reasons for this discrepancy are
discussed in Section 3.6. Of the 8 world regions in NWED, the U.S. is a net virtual water
exporter to each region, indicated by the negative virtual water balance (Table 4). The U.S. has
the largest negative virtual water balance with Eastern Asian (-2,081 $Mm^3$) and Mexico (-1,215
$Mm^3$). The U.S. is a net importer of virtual water from Central and South America (Rest of
Americas) and Europe.
Virtual water export from the U.S. is mostly agricultural commodities, such as corn,
wheat, alfalfa, for which the U.S. is a net exporter (Marston et al., 2015;Hoekstra and
Wiedmann, 2014) and mining products, such as metallic and non-metallic ores. Major virtual
water exporting regions are the Central Valley of California; the deserts of California and
Arizona; the High Plains, including the Ogallala Aquifer Region, the Arkansas River Basin, and
the Platte River Basin; the Columbia River Basin in the Pacific Northwest; Central Nevada; the
Texas Gulf Coast; the Upper Missouri River Basin in Montana; Central and Southern Florida;
and the Mississippi Embayment (Fig. 3). Many of these areas are major sources of virtual water
domestically within the U.S.; however, these results show that some areas such as southwestern
Idaho, Wyoming, and central Utah and New Mexico operate primarily in the domestic market,
and other regions such as central Nevada (metallic ores) and western Washington (non-metallic
ores) are more prominent in the international market.
The majority of virtual water exports from the United States flow through ports along the
Gulf Coast (Houston, New Orleans, Corpus Christi, Beaumont) and the West Coast (Los
Angeles/Long Beach, Washington State, San Francisco, Seattle, Portland). The ports of Los
Angeles and New York City receive the highest volume of virtual water imports followed by
Houston and Detroit. Due to where goods for export are sourced within the U.S., a world region
(or country) may receive a higher proportion of virtual water that originated as surface water or
groundwater. For example, virtual water flows through ports in the Houston metropolitan area
are dominated by groundwater sources in the Ogallala Aquifer Region, the Mississippi
Embayment aquifer system, and to a lesser extent the Central Valley of California, local
groundwater sources, and southern Arizona (Fig. 4). Mexico, Africa, and Southwest and Central
Asia are the only world regions that received more virtual water in that originated as
groundwater (Table 5; Fig 5); suggesting that exports to these regions are potentially vulnerable
to unsustainable, long-term groundwater management in the U.S. than annual fluctuations in
surface water availability and drought (Marston et al., 2015).

While we do not address surface or ground water sustainability, vulnerability, or

overdraft specifically in this paper, it is certainly desirable to combine these results with
quantification of water storage and water availability, for the purpose of policy analysis.
Conversely, Canada, Latin America, Europe, and Asia and Oceania have more exposure to
surface water fluctuations and drought but are less exposed to unsustainable groundwater
management in the U.S. Given that the U.S. is a large hydrologically, agronomically, and
climatically diverse country, it is not surprising that the type of water, surface water or
groundwater, which an international trading partner may depend on varies based on which part
of the U.S. is accessed and thus potentially causing two trading partners to have vastly different
virtual water risk profiles.

**3.4 Structural and Spatial Differences in Economic Sector Water Footprints**

The U.S. water footprint is predominantly determined by the production, manufacture,

and distribution of food. The agriculture (154,349 $Mm^3$) and livestock (15,917 $Mm^3$) economic
sectors comprise 93 % of the U.S. water footprint (181,966 $Mm^3$), with the agriculture economic
sector alone comprising 87 % of the U.S. water footprint. Overall, the agriculture and livestock
water footprint is concentrated in the Western U.S., where there is a heavy dependence on
irrigated agriculture to raise crops for human and animal consumption.

For agriculture, the Central Valley of California, the Front Range of Colorado, Central

and Southern Arizona, and the Snake/Columbia River Valley are significant geographic regions
where food is grown and where irrigation is a requisite for growing crops (Fig. 6a). Where
irrigated agriculture is not as prevalent, urban centers are moderate water footprints as they serve
as regional distribution for food (Omaha, Nebraska; Wichita, Kansas; Dallas, Houston, and
Brownsville, Texas; New Orleans, Louisiana; Northwest Arkansas; and Central Florida). The
U.S. livestock footprint is more concentrated on the west coast U.S. and Snake River Valley of
Idaho; however, on the east coast, the Carolinas have the largest livestock water footprint (Fig.
6c). Outside these areas, the U.S. livestock water footprint is concentrated around cities where
there is a relatively large inflow of virtual water with little to no virtual water outflows.

Unlike the U.S. water footprint of agriculture and livestock, in which both rural and

urban counties play significant roles, the U.S. industrial water footprint (Fig. 6b), and to the same
extent the U.S. water footprint of and power production and flow and domestic water
consumption (Fig. 6e and 6f), is dominated by urban areas. Not surprisingly, domestic and
industrial water use is highly co-located with urban areas as are virtual water inflows and
outflows. Major nodes in the U.S. industrial water footprint network are Chicago, Illinois;
Houston and Dallas, Texas; Los Angeles California; Seattle, Washington; Phoenix, Arizona; Las
Vegas, Nevada; the Boston-Washington Corridor; Central and Southern Florida; and each major
metropolitan area east of the Mississippi River. While the same areas are important in the
domestic water footprint, the U.S. southwest – Southern California, Central and Southern
Arizona, and Las Vegas, Nevada – have the largest domestic water footprints.

The U.S. mining water footprint is highly dependent on the location of mineral resources

in addition to processing facilities and distribution hubs. Some geographic regions with
substantial mining water footprint do not have a significant water footprint in other sectors; for
example, northern Alaska; west Texas; the Gulf Coast; Oklahoma; North Dakota; northern
Michigan and Minnesota; and parts of Nevada, Montana, Utah, New Mexico, and Wyoming
(Fig. 6d). Southern California, and to a lesser extent Southern Arizona, is an exception to this
because these are regions with substantial mining activity – oil and gas in Southern California
and hard rock mining in Arizona – that are co-located with agricultural and industrial production
in addition to high domestic water consumption.

The net export status of a county matters because a net virtual water exporter may have a

very different approach to national water policy discussions than a net importer (Fig. 7). The
(usually medium-sized) communities that sit in between the net-importing and net-exporting
categories may take a distinct and more balanced position on national policy. Agricultural
western communities tend to be net exporters, urban communities tend to be net importers, and
rural eastern communities tend to be relatively neutral; midsize urban communities, such as those
commonly found in the Midwest and East, may be relatively neutral as well.

**3.5 Uncertainty Introduced by Consumption Coefficient Estimates**

At the county-level, blue water footprint uncertainties introduced by consumption

coefficients range several orders of magnitude in $Mm^3$ and relative percent (Fig. 8). The small
rural counties of Bristol Bay Borough, Alaska and Kenedy County, Texas have the smallest
water footprint uncertainties (<0.50 $Mm^3$). Los Angeles County, California has the largest water
footprint uncertainty (4,050 $Mm^3$). After Los Angeles, 3 counties have a water footprint
uncertainty between 3,000 – 4,000 $Mm^3$; 7 counties have a water footprint uncertainty between
2,000 – 3,000 $Mm^3$; 42 counties have a water footprint uncertainty between 1,000 – 2,000 $Mm^3$;
and 79 counties have a water footprint uncertainty between 500 – 1,000 $Mm^3$. In relative terms,
county-level water footprint uncertainty is 58.2 % – 99.9 % of a county's total water
withdrawals. Relative water footprint variation tends to increase in the Eastern United States.
However, in absolute terms, consumption coefficient variation is more important in the western
U.S. due to the potentially large variation in virtual water outflows from the U.S.'s largest virtual
water sources.
A community's role in the hydro-economic network, and its perspective on hydro-economic
policy issues, can qualitatively change depending on our uncertainty. Uncertainties introduced by
the consumption coefficients, which are quite large in absolute terms, roughly 17 % of U.S.
counties can switch between roles as a net virtual water importer and exporter (+ or - $VW_{Balance}$)
depending on the consumptive use assumptions (Fig. 9).
Results using the withdrawal-based ($CU = 1$) scenario are located in the Supplemental
Information (Table SI 4-D).

**3.6 Uncertainty in International Virtual Water Flow**
As mentioned in Section 3.4, there are several potential reasons for the discrepancy in the
magnitude of virtual water flows. First, there are differences in the underlying source data for
international trade and water use. NWED utilizes commodity flows modeled by FAF, which
itself utilizes Census Foreign Trade Data for 2010 (Southworth et al., 2010;Hwang et al., 2016),
while benchmark international virtual water trade studies utilized trade data from the
International Trade Centre averaged between 1996-2005 (Water Footprint Network, 2013).
Additionally, the source water data for the U.S. are different. NWED utilizes USGS water
withdrawal data, which is self-reported with state-level variations (Marston et al., 2018; Maupin
et al., 2014), benchmark international virtual water trade studies utilized CROPWAT modeling
(Water Footprint Network, 2013). Secondly, despite controlling for port influences, it is likely
that more virtual water is attributed to ports than necessary, which would dampen international
virtual water flows in NWED. NWED has difficulty handling 'flow through' virtual waters flow
that would be otherwise assigned to a point of final consumption. In this case, a flow through
entity may be assigned virtual water flow at the port or another distribution hub.  Lastly, previous
international virtual water studies included the water use of inputs in the virtual water flow of a
commodity, e.g., the water consumption for animal feed as part of animal products related virtual
water flow. A method to handle this is under development for the next version of NWED. While
there are disadvantages to the current method in which international trade is modeled in NWED,
methods to improve this aspect of the data product are ongoing and there is data structure in
place to merge additional international trade flow datasets with the current NWED data structure.

**3.7 Temporal Uncertainty**

As mentioned previously, the NWED data are limited in representativeness to roughly the

2010 – 2012 post-recession timeframe but are not precisely linked to a single year. Temporal
uncertainty is introduced by utilizing annual timescale data. Given this, NWED data are more
directly relevant to surface water management than to groundwater management because surface
water has months to a few years of storage, and groundwater has centuries of storage, but in the
future we could use this data to analyze sustainability and vulnerability of water usage.

**4.  Conclusions**

Mekonnen and Hoekstra reported that the U.S. combined blue and grey water footprint,

which is the closest equivalent to the water sources used NWED, to be 320,496 Mm$^3$ and 874 m³
capita$^{-1}$ (Mekonnen and Hoekstra, 2011a). Results from NWED, which uses 4 consumptive use
scenarios, for the median annual water footprint, $F_{CUMed}$, of the U.S. is 181,966 Mm³ ($F_{Withdrawal}$:
400,844 Mm³; $F_{CUMax}$: 222,144 Mm³; $F_{CUMin}$: 61,117 Mm³). On a per-capita basis, results from
NWED found the median U.S. water footprint ($F'_{CUMed}$) is 589 m³ capita$^{-1}$ ($F'_{Withdrawal\text{-}Basis}$: 1298
m³ capita$^{-1}$; $F'_{CUMax}$: 720 m³ capita$^{-1}$; $F'_{CUMin}$: 198 m³ capita$^{-1}$). Given these statistics, the reported
Mekonnon and Hoekstra water footprint and per capita water footprint falls between the
*withdrawal-based (CU=1)* and maximum consumptive use coefficient ($CU_{Max}$) scenarios.
Depending on the assumptions about consumptive use at the economic-sector level, these two
datasets are in rough agreement regarding the magnitude of the U.S. water footprint.

The uncertainty introduced by water use data and consumption coefficients demonstrate

the great need for the development of region-specific, sector-level water use data and
consumption coefficients for the entire U.S. For example, water footprint uncertainty is roughly
58 % to over 99 % of a county's total water footprint, which increases in the eastern United
States. However, in absolute terms, consumption coefficient variation is more important in the
western U.S. due to the potentially large variation in virtual water outflows from the agricultural
sector with largest blue water withdrawals. While we have presented results for the $CU_{Med}$
scenario in this paper, we must recognize the potentially large variation in water consumption
that could exist compared to what is reported. Therefore, conclusions drawn from NWED data,
as well as those drawn from the underlying water data, must recognize the large range of
uncertainty with respect to water withdrawal and consumption in the U.S. Nevertheless, there are
still general observable trends in U.S. virtual water flows and water footprints, which are
presented below.

The U.S. hydro-economic network is centered on cities and is dominated by the local and

regional scales of trade, with medium-sized cities playing a disproportionate role. The proper
framing of water governance and policy may be proportional to the structure of that network.
Large cities source from all sizes of communities, but small and rural communities mostly source
from other small communities, leading to a structural difference between the diversity and
connectivity of urban and rural water supply chains. Further, medium-size metropolitan areas
have a unique role in the U.S. hydro-economic as the link between rural virtual water production
and urban virtual water consumption and are the most important single scale of community in the
network. The U.S. hydro-economic network's connections and power structures are primarily
local and regional except for the large metropolitan areas that operate at the national level and
large-city ports that operate at the international level. This scale-specific finding is novel because
most prior work on water footprints focuses on international trade.

Within the U.S., urban counties have a strong hydro-economic dependence on rural

counties: for the $CU_{Med}$ scenario, there is a virtual water transfer of 114,953 Mm$^3$ from rural
counties to urban counties, roughly a third of all virtual water flow in the U.S., with only a
33,876 Mm$^3$ return flow of virtual water. However, there is also strong urban-to-urban hydro-
economic dependence. The virtual water transfer between urban counties is of the same
magnitude as the rural-to-urban virtual water transfers (111,458 Mm$^3$). Taken together, rural-to-
urban and urban-to-urban virtual water flow accounts for approximately 58 % of U.S. domestic
virtual water flow, illustrating the urban demand for not just water-intensive food sourced from
rural counties, but also water-intensive power and industrial products sourced from urban
counties. Further work on characterizing county-level virtual water flows can extend the logic
developed by frameworks to characterize catchment-level water use regimes (Weiskel et al.,
2007) to hydro-economic networks. Specifically, NWED data can provide a socio-hydrological
extension to previous work on hydroclimatic regime classification in the U.S. (Weiskel et al.,

2014).

The networked structure of water footprint sources creates systemic exposure to surface

water scarcity and groundwater unsustainability at virtual water source locations. The U.S. and
the global economy are particularly exposed to drought, and other system shocks, in the Western
U.S. generally, especially in California, Central and Southern Arizona, Idaho, and the Great
Plains. In the Eastern U.S., exposure to drought, or other system shocks, presents in South Texas,
South Florida, the Chicago area, and the Lower Mississippi Valley. Because the whole U.S., and
world, depend on these water supplies, these locations should be a priority for national water
policy (Cooley and Gleick, 2012; Gleick et al., 2012); for public investment in water
infrastructure to manage drought (Brown and Lall, 2006; Galloway Jr, 2011); and for innovative
green infrastructure and market-based solutions that address water supply and demand problems.
Additionally, the ports through which virtual water flows create transportation risks posed by
war, strikes, tropical storms, earthquakes, and sea level rise. These locations should be a priority
for national resilience policies and efforts, and alternative freight corridors should be developed
so that port closures do not impact the ability of U.S. businesses to get their water-intensive
goods to domestic and international markets (or vice versa).

Given the networked structure of the FEW system, the strong urban-rural dependence of

FEW system flows, and the uncertainties presented by information gaps, future FEW system
studies must address questions of worldview. For example, questions regarding which scale is
the right scale (Vörösmarty et al., 2010;Vörösmarty et al., 2015) and which decision boundary is
the best decision boundary (Rushforth et al., 2013) for understanding the FEW system
interactions are dependent on the worldview of stakeholders and policymakers. In the U.S., the
direct and indirect transfer of FEW system resources is concentrated at the mesoscale – regions
and/or county equivalents – and not the national or global scales. This has implications for
developing robust FEW system policy: the mesoscale is a manageable scale and there is the
ability to manage aspects of FEW systems and craft FEW system interventions at this scale
through extant and novel local and regional governance systems. For example, downstream-
driven, market-based supply chain governance of "soft" supply chains by major retailers and
distributors; downstream-driven City-driven governance via their hard infrastructures
(McManamay et al., 2017); upstream-driven, watershed- or river-driven governance wherein
infrastructure managers consider how the services of their water propagate through the economy;
or FEW governance where F, E, and W agents work together because these sectors have the
largest footprints.

NWED provides insight into which sectors and geographic areas need to be prioritized in

the development of these consumption coefficients. The lack of certainty on consumption
coefficients (*Section 3.5*) limits the ability to estimate or gauge one area's exposure to
hydrological hazards in another area in its supply chain and must be addressed through the
development of county- or region-specific and economic sector-specific consumption
coefficients. We suggest starting with cities and irrigated agriculture in the Western U.S. due to
the major influence that consumption coefficients have on water footprints, and because we lack
locally accurate consumption coefficients to distinguish between regions this prevents us from
accurately assessing local water balances or scarcity.

Despite basic limitations imposed by the primary data sources, NWED is a robustly

quantified blue water footprint; future refinements to NWED will seek to address these
limitations and add additional functionality, such increased resolution on pass-through
commodity flows. The empirical basis of this analysis, along with its economic completeness
and spatial detail, make this result a landmark resource in the scientific discussion of water
footprints, virtual water flow, and the sustainability and resilience of a nation's water resources
in the connected global economy.
**Code Availability:**
The NWED 1.1 code will be made available on GitHub:  https://github.com/NWED/v1.1.
**Data Availability:**
NWED version 1.1 is available at the Hydroshare data repository and can be accessed at:
https://www.hydroshare.org/resource/84d1b8b60f274ba4be155881129561a9/
**Appendices:**
**Appendix 1: Difference Between NWED Version 1.0 and 1.1**
Data from NWED 1.0 have previously been published in by Rushforth and Ruddell
(Rushforth and Ruddell, 2016). While the methodology is largely the same, there are key
differences between the two versions of NWED.
- If updated disaggregation and attraction factors were available, these factors were
updated.
- Specifically, agricultural disaggregation factors were updated at the crop level
using the latest USDA NASS.
- Additionally, the mining sector been updated to have commodity code specific
disaggregation factors using the location of mines and mineral production as
disaggregation factors rather than employment.
- The power sector and domestic sector has been added to NWED version 1.1.
- Export virtual water flows have been disaggregated from virtual water flows to
port cities.
- Import virtual water flows have been added to NWED version 1.1.
• The $CU_{Max}$, $CU_{Med}$, and $CU_{Min}$ consumption scenarios were added to NWED
version 1.1.
• Groundwater and surface water disaggregation of virtual water flows for
withdrawal, $CU_{Max}$, $CU_{Med}$, and $CU_{Min}$ scenarios were added.

**Appendix 2: NWED Glossary**
*Agricultural Sector*: NWED sector comprised of farm-based activities to grow crops for food or
industrial purposes. Irrigation is the primary water using activity in the agricultural sector
(Maupin et al., 2014).
*Attraction Factor*: A fraction used to disaggregate commodity flows on the consumption side. In
NWED 1.1, population is used as an attraction factor. Each county within a FAZ is assigned a
fraction equivalent to its percent of the total population.
*County*: A county or county equivalent (parish, borough, Washington D.C., or a independent
city) is a sub-state geographic scale that is roughly equivalent to the mesoscale.
*Destination*: The geographic location where a commodity flow terminates.
*Freight Analysis Zone (FAZ)*: A group of counties that represents a metropolitan statistical area,
census statistical area, or remainder of state (Southworth et al., 2010; Hwang et al., 2016)
*Industrial Sector*: Economic sector that produces industrial goods. Water use in the industrial
sector includes, "fabricating, processing, washing, diluting, cooling, or transporting a product;
incorporating water into a product; or for sanitation needs within the manufacturing facility,"
(Maupin et al., 2014).
*Large Central Metro Counties*: U.S. counties with greater than 1 million inhabitants that are the
central county of a metropolitan statistical area (Ingram and Franco, 2012).
*Large Fringe Counties*: U.S. counties with greater than 1 million inhabitants that are not the
central county of a metropolitan statistical area (Ingram and Franco, 2012).
Livestock Sector: Economic sector comprised of the raising of animals for animal products in
addition to aquaculture activities. Water use in the livestock sector only includes direct water use
at livestock, and related facilities (Maupin et al., 2014).
*Medium Metro Counties*: U.S. counties with between 250,000 and 999,999 inhabitants (Ingram
and Franco, 2012).
*Micropolitan Counties*: U.S. counties with between 10,000 and 49,999 inhabitants that have an
urban cluster (Ingram and Franco, 2012).

*Mining Sector*: Economic sector comprised of mineral producing activities, including metallic
and non-metallic ore, in addition to sand and gravel, crude petroleum and natural gas. Water
using activities in the mining sector include, "Mining water use is water used for the extraction
of minerals that may be in the form of solids, such as coal, iron, sand, and gravel; liquids, such as
crude petroleum; and gases, such as natural gas," (Maupin et al., 2014).
*Non-Core Counties:* U.S. counties with between 10,000 and 49,999 inhabitants that do not have
an urban cluster (Ingram and Franco, 2012).
*Origin*: The geographic location where a commodity flow originates.
*Production Factor*: A fraction used to disaggregate commodity flows on the production side. In
NWED 1.1, multiple production factors are used specific to the economic sector. Each county
within a FAZ is assigned a fraction equivalent to its percent of the total population.
*Power Sector*: NWED sector comprised of electric generating stations, which includes
thermoelectric and non-thermoelectric facilities (renewable energy sources). Water is used at
thermoelectric generation stations in addition to hydroelectric facilities.
*Small Metro Counties*: U.S. counties with metropolitan statistical areas with less than 250,000
inhabitants (Ingram and Franco, 2012).
*Virtual Water*: Also known as indirect water or embodied water, has been studied as a strategic
resource for two decades as it allows geographic areas (country, state, province, city) to access
more water than is physically available (Allan, 1998; Allan, 2003; Suweis et al., 2011; Dalin et
al., 2012; Dang et al., 2015; Zhao et al., 2015; Marston et al., 2015).
*Virtual Water Inflows into a Geographic Area ($VW_{In}$)*: The volume of water indirectly consumed
to produce goods or services produced outside a geographic boundary of interest for
consumption within that geographic boundary of interest.
*Virtual Water Outflows from a Geographic Area ($VW_{Out}$)*: The volume of water used to produce
goods or services that are consumed outside of geographic boundary of interest.
*Virtual Water Balance of a Geographic Area ($VW_{Net}$)*: Virtual water Inflows minus virtual water
outflows for a geographic boundary of interest.
*Water Footprint*: the volume of surface water and groundwater consumed during the production
of a good or service and is also called the virtual water content of a good or service (Mekonnen
and Hoekstra, 2011b).
*Water Footprint of Consumption*: water consumption for local use in addition virtual water
import (Mekonnen and Hoekstra, 2011a)
*Water Footprint of a Geographic Area (F)*: The volume of water representing direct water
consumption plus virtual water inflows minus virtual water outflows for a geographic boundary
of interest. A per-capita water footprint ($F\grave{}$) is F divided by the population within the geographic
boundary of interest.

*Water Footprint of Production*:  the total volume of water consumed with a geographic
boundary, including water consumption for local use less virtual water export (Mekonnen and
Hoekstra, 2011a).

*Water Consumption (C)*:  The total volume of water consumed from a water source, when
consumption is withdrawals minus return flows. A water source is either surface water or
groundwater. NWED utilizes four consumptive use scenarios based on a withdrawal-based
scenario, and minimum, median, and maximum consumptive use scenario. Consumptive use
scenarios are based on reports published by the United States Geological Survey (Shaffer and
Runkle, 2007).

*Water Withdrawal (W)*:  The total volume of water withdrawn from a water source. A water
source is either surface water or groundwater.


**Appendix 3: Commodity Trade Linkage Metrics**
Each commodity trade linkage is measured by 15 metrics: $-t$, $\$$, $tm$, $VW_{I_n,J_m,c,s,t,k,W_{Total}}$,
$VW_{I_n,J_m,c,s,t,k,W_{SW}}$, $VW_{I_n,J_m,c,s,t,k,W_{GW}}$, $VW_{I_n,J_m,c,s,t,k,CU_{Max,Total}}$, $VW_{I_n,J_m,c,s,t,k,CU_{Max,SW}}$,
$VW_{I_n,J_m,c,s,t,k,CU_{Max,GW}}$, $VW_{I_n,J_m,c,s,t,k,CU_{Med,Total}}$, $VW_{I_n,J_m,c,s,t,k,CU_{Med,SW}}$, $VW_{I_n,J_m,c,s,t,k,CU_{Med,GW}}$,
$VW_{I_n,J_m,c,s,t,k,CU_{Min,Total}}$, $VW_{I_n,J_m,c,s,t,k,CU_{Min,SW}}$, $VW_{I_n,J_m,c,s,t,k,CU_{Min,GW}}$.
**Team List:**
Richard R Rushforth
Benjamin L. Ruddell

**Author Contribution:**
R. Rushforth developed the NWED methodology and the executed code to carry out the
methodology. R. Rushforth wrote the manuscript with help from B. Ruddell.

**Competing Interests:**

The authors declare that they have no conflicts of interest.

**Disclaimer:**


The opinions expressed by authors contributing to this journal do not necessarily reflect
the opinions of the Hydrology and Earth System Sciences Journal or the institutions with which
the authors are affiliated.

**Acknowledgements:**
Funding for this research was provided by the National Science Foundation under award
number ACI-1639529 (FEWSION). The opinions expressed are those of the authors, and not
necessarily the National Science Foundation. The authors would like to acknowledge input from
colleagues on the development of this manuscript and the anonymous peer referees of this paper.
Finally, the authors would like to thank the anonymous referees of this paper for their thorough
and constructive comments.

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

**Table 1. Minimum, Median, and Maximum Consumption Use Coefficients (CU) Used to Estimate Consumptive Water Use in NWED[1]**

| Sector ($s$) | $CU_{Min}$ | $CU_{Med}$ | $CU_{Max}$ | $N^2$ |
|---|---|---|---|---|
| Irrigated Agriculture | 37 % | 100 % | 100 % | 170 |
| Domestic | 0 % | 13 % | 73 % | 229 |
| Industrial | 0 % | 10 % | 35 % | 219 |
| Livestock | 10 % | 100 % | 100 % | 158 |
| Mining | 0 % | 14 % | 86 % | 141 |
| Power | 0 % | 2 % | 75 % | 216 |

[1]Consumption coefficients adapted from (Shaffer and Runkle, 2007).

[2]The number of studies evaluated to approximate the consumption coefficients.

**Table 2. U.S. Water Footprint and Virtual Water Statistics**

| Virtual Water Statistic | Withdrawal-Based ($CU = 1$) | $CU_{Max}$ | $CU_{Med}$ | $CU_{Min}$ |
|---|---|---|---|---|
| Water Use – Domestic ($Mm^3$) | 37,566 | 27,423 | 4,884 | 0 |
| Water Use – Non-Domestic ($Mm^3$) | 366,687 | 200,712 | 181,773 | 60,722 |
| Water Use – Total ($Mm^3$) | 404,253 | 228,135 | 186,657 | 60,722 |
| Virtual Water Outflows, $VW_{Out}$ ($Mm^3$) | 362,690 | 196,857 | 178,622 | 59,870 |
| Virtual Water Inflows, $VW_{In}$ ($Mm^3$) | 359,282 | 190,866 | 173,931 | 60,265 |
| Virtual Water Balance, $VW_{Bal}$ ($Mm^3$) | -3,409 | -5,991 | -4,691 | 395 |
| Virtual Water Export, $VW_{Export}$ ($Mm^3$) | 10,671 | 9,039 | 7,739 | 2,653 |
| Virtual Water Import, $VW_{Import}$ ($Mm^3$) | 7,263 | 3,048 | 3,048 | 3,048 |
| Non-Domestic Water Footprint ($Mm^3$) | 363,279 | 194,722 | 177,082 | 61,117 |
| Total Water Footprint ($Mm^3$) | 400,844 | 222,144 | 181,966 | 61,117 |
| Total Water Footprint Per Capita ($m^3$ capita$^{-1}$) | 1,298 | 720 | 589 | 198 |
| Central Water Footprint Per Capita ($m^3$ capita$^{-1}$) | 828 | 399 | 282 | 97 |
| Fringe Water Footprint Per Capita ($m^3$ capita$^{-1}$) | 981 | 368 | 250 | 83 |
| Medium Water Footprint Per Capita ($m^3$ capita$^{-1}$) | 1,705 | 1,076 | 936 | 315 |
| Small Water Footprint Per Capita ($m^3$ capita$^{-1}$) | 1,794 | 1,139 | 992 | 333 |
| Micro Water Footprint Per Capita ($m^3$ capita$^{-1}$) | 1,876 | 1,169 | 1,024 | 345 |
| Non-Core Water Footprint Per Capita ($m^3$ capita$^{-1}$) | 1,927 | 1,217 | 1,053 | 344 |
| Rural to Urban VW Transfers ($Mm^3$) | 114,953 | 70,648 | 66,524 | 22,496 |
| Rural to Rural VW Transfers ($Mm^3$) | 91,682 | 63,698 | 60,676 | 20,614 |
| Urban to Urban VW Transfers ($Mm^3$) | 111,458 | 39,921 | 32,338 | 10,459 |
| Urban to Rural VW Transfers ($Mm^3$) | 33,876 | 13,551 | 11,345 | 3,647 |

**Table 3. Blue Virtual Water Transfers Between Urban and Rural Areas ($Mm^3$)**

| Urban/Rural Classification (↑ Urban Rural ↓) | ← Urban | | | | | Rural → | | |
|---|---|---|---|---|---|---|---|---|
| | Central | Fringe | Medium | Small | Micro | Non-Core | $VW_{Out, CUMed}$ | $VW_{Balance, CUMed}$ |
| Central | 2,529 | 628 | 593 | 201 | 139 | 72 | 4,162 | 19,299 |
| Fringe | 2,644 | 1,632 | 1,477 | 505 | 447 | 306 | 7,011 | 9,779 |
| Medium | **5,345** | 3,174 | 14,316 | 4,311 | 3,371 | 1,992 | 32,510 | 26,102 |
| Small | 4,022 | 2,318 | 8,626 | 4,111 | 3,607 | 2,138 | 24,822 | 2,757 |
| Micro | 3,821 | 3,812 | 14,153 | 7,710 | 8,302 | 4,837 | 42,634 | -15,755 |
| Non-Core | 5,100 | **5,227** | **19,446** | **10,740** | **11,013** | **8,218** | 59,744 | -42,182 |
| $VW_{In, CUMed}$ | 23,460 | 16,790 | 58,612 | 27,579 | 26,879 | 17,562 | 170,883 | – |

13 **Table 4. Urban-Rural Blue Virtual Water Transfer by Economic Sector (Mm³)**

| Origin County | Destination County | Sector | Virtual Water Flow (Mm³) |
|---|---|---|---|
| Urban | Urban | Power | 22 |
| Urban | Urban | Agriculture | 27,743 |
| Urban | Urban | Industrial | 2,694 |
| Urban | Urban | Livestock | 1,714 |
| Urban | Urban | Mining | 165 |
| Urban | Rural | Power | 6 |
| Urban | Rural | Agriculture | 9,583 |
| Urban | Rural | Industrial | 733 |
| Urban | Rural | Livestock | 950 |
| Urban | Rural | Mining | 73 |
| Rural | Urban | Power | 13 |
| Rural | Urban | Agriculture | 59,119 |
| Rural | Urban | Industrial | 955 |
| Rural | Urban | Livestock | 6,100 |
| Rural | Urban | Mining | 337 |
| Rural | Rural | Power | 159 |
| Rural | Rural | Agriculture | 53,731 |
| Rural | Rural | Industrial | 848 |
| Rural | Rural | Livestock | 5,764 |
| Rural | Rural | Mining | 175 |
| Urban | Urban | Domestic | 3,715 |
| Rural | Rural | Domestic | 1,168 |

     **Table 5. U.S. Blue Virtual Water Exports and Imports to and Balances with World Regions**

| Region | Virtual Water Export (Mm$^3$) | % SW | % GW | Virtual Water Import (Mm$^3$) | % SW | % GW | Virtual Water Balance (Mm$^3$) |
|---|---|---|---|---|---|---|---|
| Canada | 1,078 | 51% | 49% | 973 | — | — | -105 |
| Mexico | 1,787 | 40% | 60% | 572 | — | — | -1,215 |
| Rest of Americas | 672 | 67% | 33% | 597 | — | — | -75 |
| Europe | 662 | 53% | 47% | 266 | — | — | -396 |
| Africa | 448 | 33% | 67% | 43 | — | — | -405 |
| Southwest & Central Asia | 355 | 45% | 55% | 102 | — | — | -253 |
| Eastern Asia | 2,307 | 62% | 38% | 226 | — | — | -2,081 |
| Southeast Asia & Oceania | 432 | 61% | 39% | 269 | — | — | -163 |
| **Total** | **7,741** | **52%** | **48%** | **3,048** | **—** | **—** | **-4,693** |

*SW – Surface Water; GW– Groundwater*

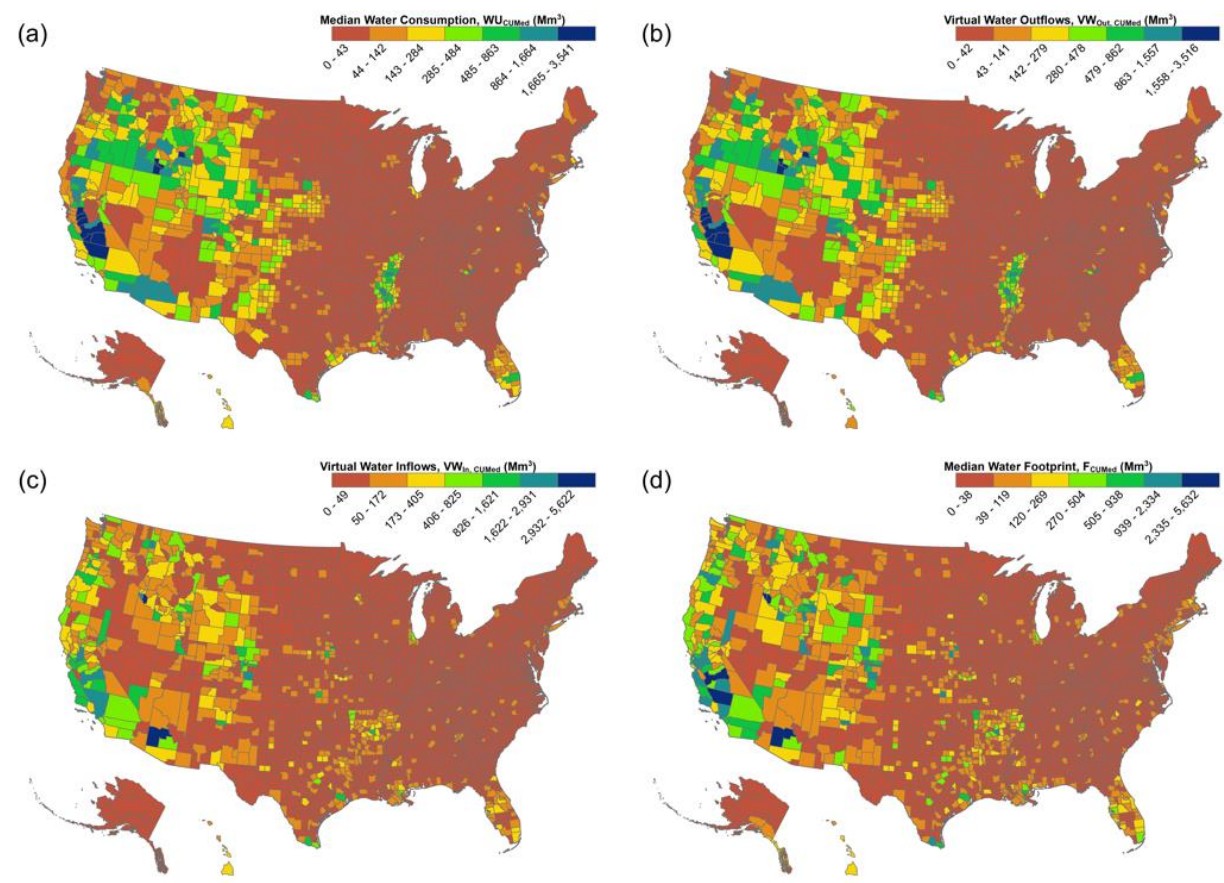

**Figure 1. (a) Median county-level blue water consumption in the U.S. (b) Blue virtual water outflows from U.S. are concentrated in the western United States, particularly where irrigated agriculture is located, in addition to the High Plans, Mississippi Embayment, and south Florida. (c) Blue virtual water inflows are concentrated in Western U.S. cities, Western U.S. agricultural counties, metropolitan regions in the Eastern U.S., and in particular where a city also serves as a regional distribution center or has prominent food processing industry (Little Rock and Northwestern Arkansas, Chicago and Houston). (d) Annual Withdrawal-Based ($CU_{Med}$) Blue Water Footprint, $F_{CUMed}$ [Mm³], for U.S. Counties.**

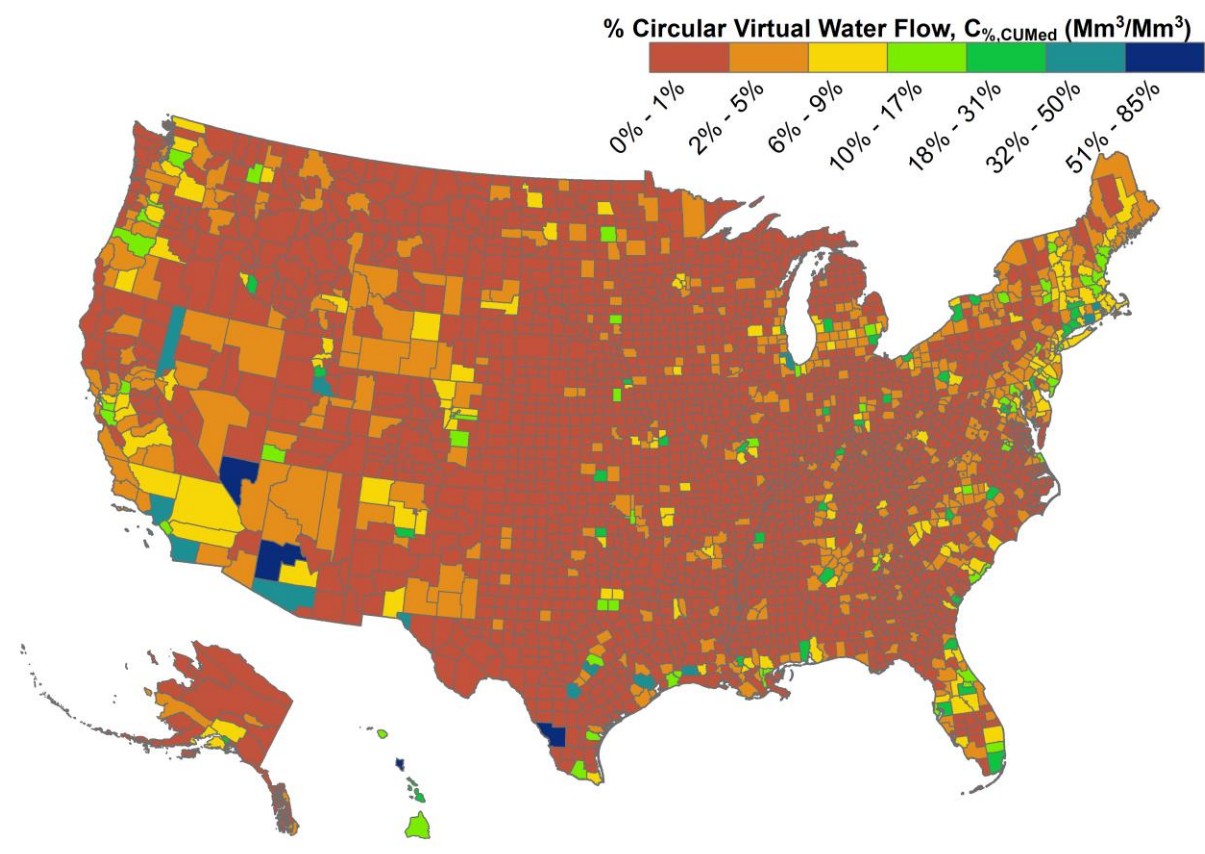

**Figure 2. Circular blue virtual water flows ($CU_{Med}$), or blue virtual water flows that**
**originate and terminate within the same county. This is a map of the use of "local water" in**
**the hydro-economy. Phoenix, Arizona is a local water hotspot.**

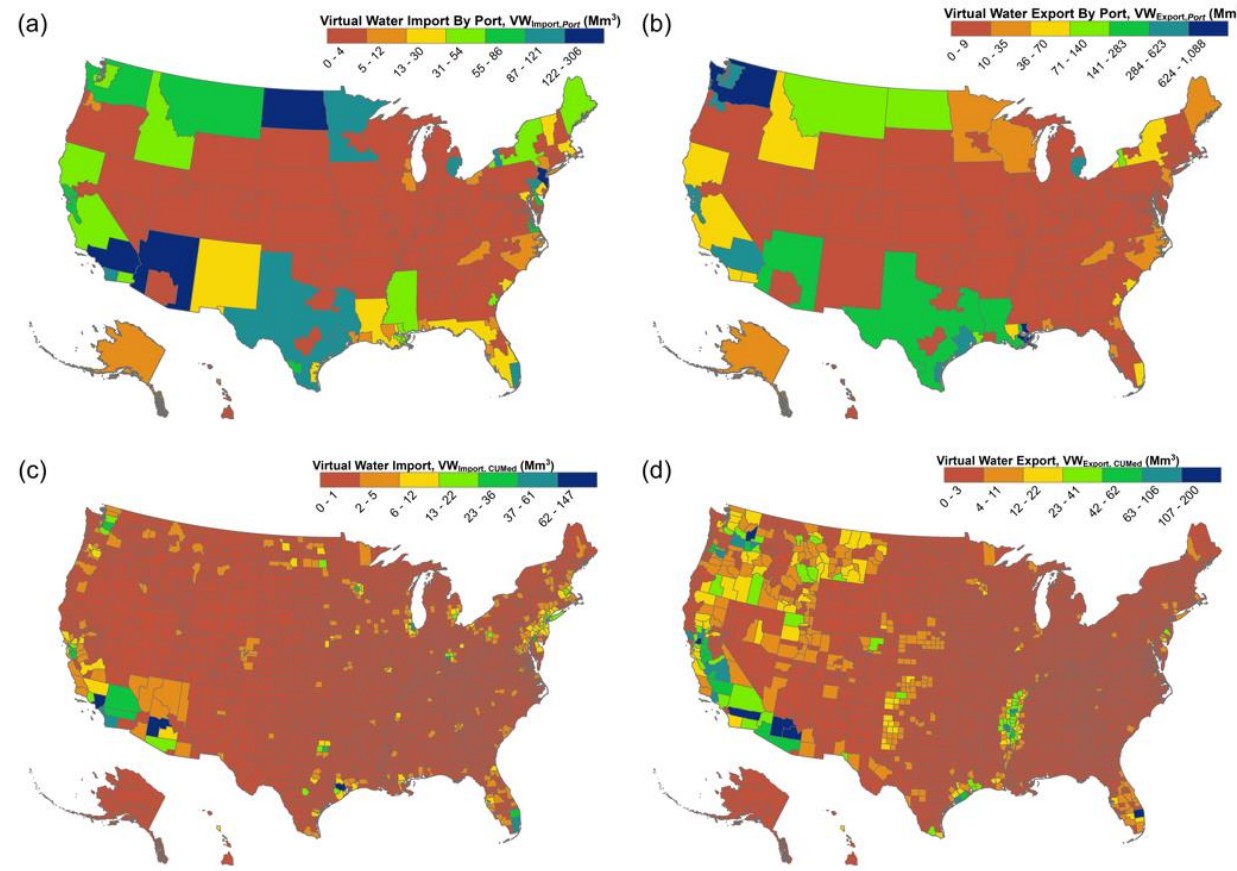

**Figure 3. (a) The port and border regions through which the majority of U.S. blue virtual water imports (*CU_Med*) enter the U.S. market are primarily Los Angeles, New York, Arizona, North Dakota, Houston, Detroit, Buffalo and Detroit (FAZ's are used for port region boundaries). However, the whole land border with Canada and Mexico is import to U.S. virtual water import. (b) The ports through which the majority of U.S. virtual water exports (*CU_Med*) enter the global market are located in natural hazard prone areas along the West Coast, Gulf Coast, and Eastern Seaboard. (c) Cities such as Los Angles, Phoenix, Houston, New York City, Miami, Dallas, Seattle, and the San Francisco Bay area are the major destinations of U.S. virtual water imports (*CU_Med*). (d) U.S. virtual water exports (*CU_Med*) originate from California's Central Valley; Southern California and Southwest Arizona; the Columbia River Basin and the Pacific Northwest; Central Nevada and Northwest Utah; the Ogallala Aquifer region of the Midwest; the Texas Gulf Coast; the Mississippi Embayment; and South Florida.**

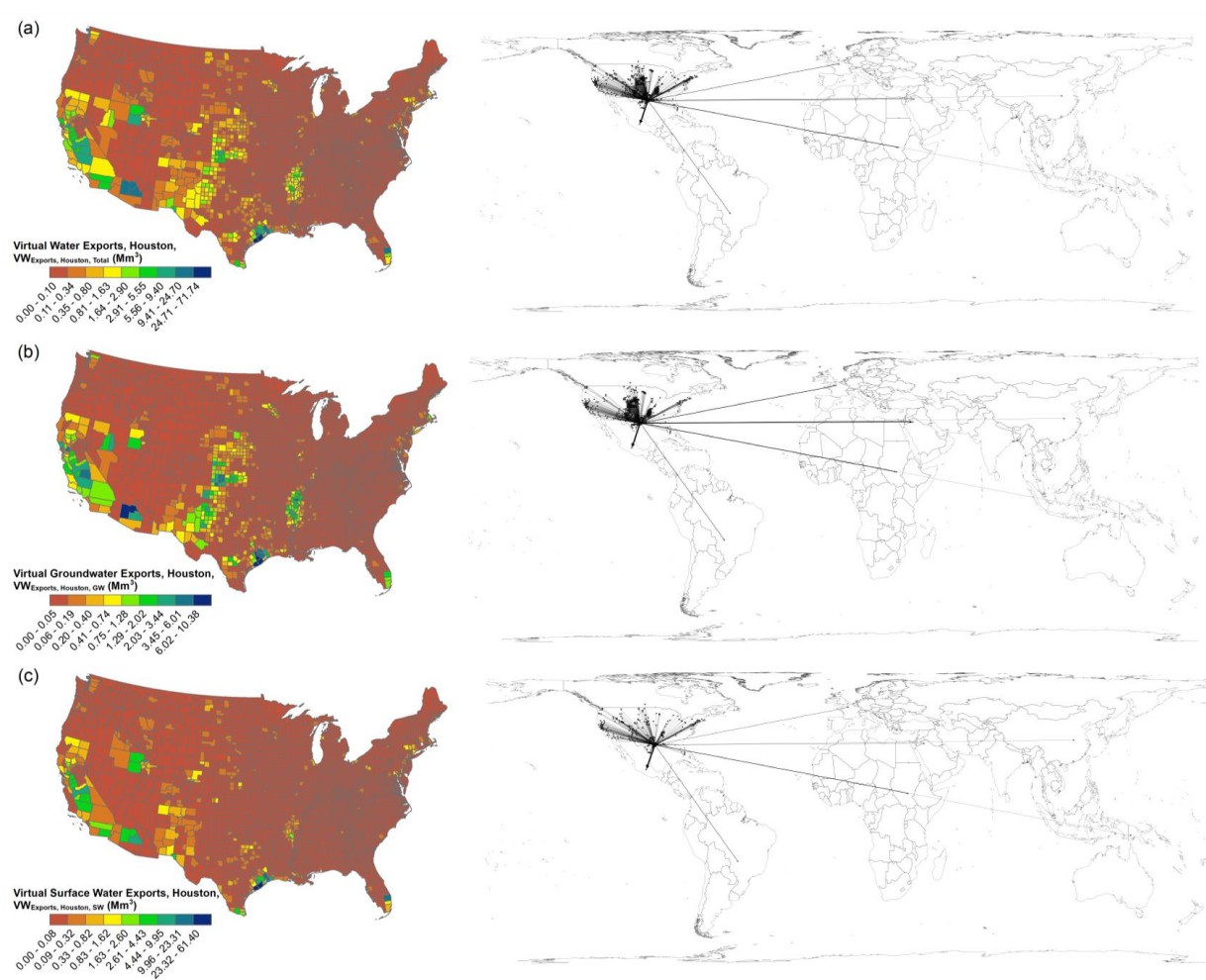

**Figure 4. (a) U.S. blue virtual water exports ($CU_{Med}$) through ports in the Houston metropolitan area are sourced from the Central Valley of California, Central Utah and Northern Utah, Southern Arizona, the Ogallala Aquifer Region, South Texas and the Texas Gulf Coast, and the Mississippi Embayment aquifer region. Virtual water flows into the Houston ports and then is redistributed to the 8 world regions in NWED. Mexico is the largest recipient of virtual water flows from Houston ports. (b) Virtual groundwater flow through Houston ports is sourced from the Central Valley of California, Central Utah and Northern Utah, Southern Arizona, the Ogallala Aquifer Region, South Texas and the Texas Gulf Coast, and the Mississippi Embayment aquifer region. (c) Virtual surface water through Houston ports is sourced from the Central Valley of California, Southern California, the Phoenix Metropolitan Area, Northern Utah, and the Texas Gulf Coast. Network maps are plotted with Gephi using the Map of Countries and GeoLayout plugins.**

65

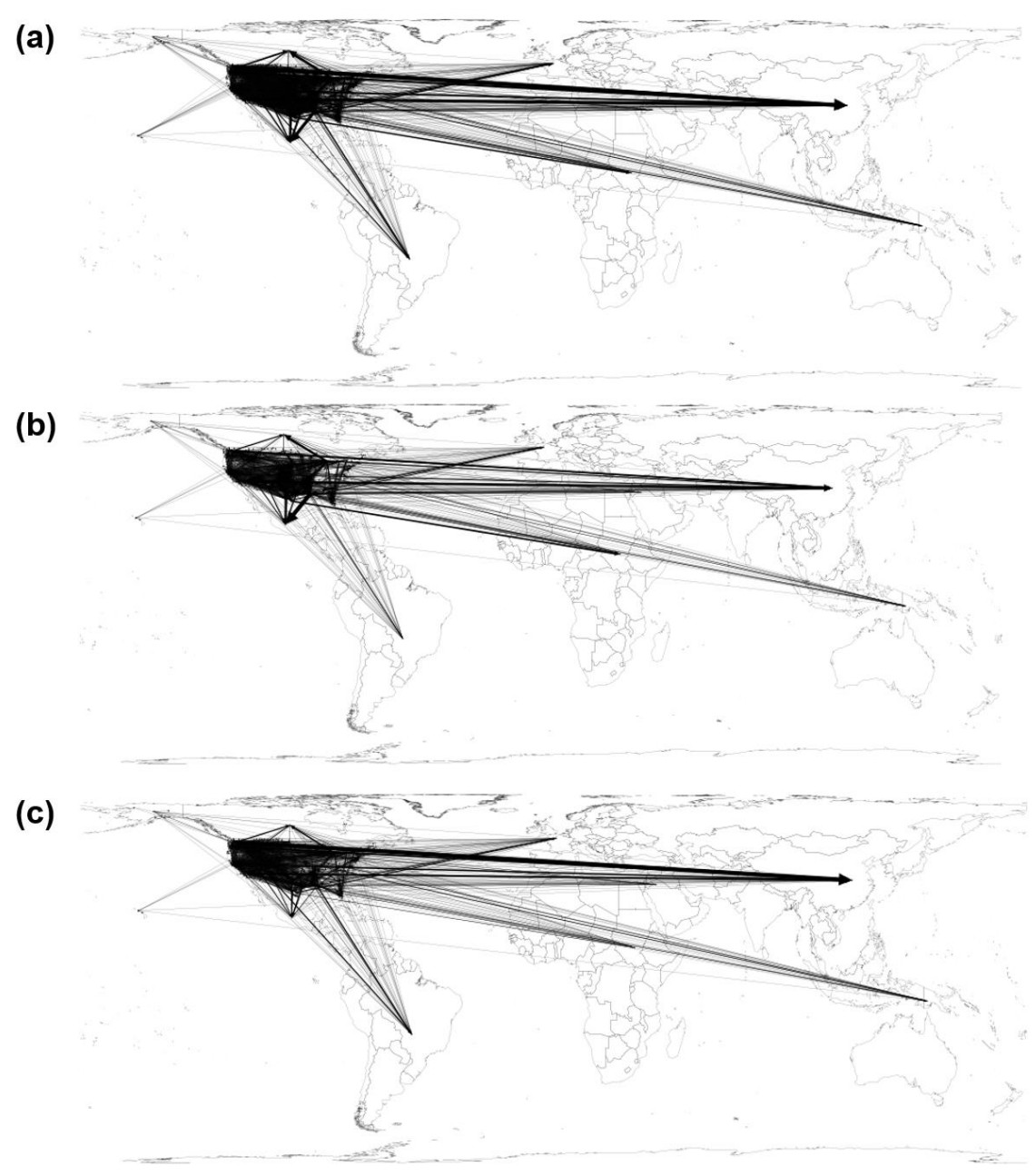

66

**Figure 5. (a) U.S. blue virtual water exports ($CU_{Med}$) through all U.S. ports. Only flows > 0.1 Mm³ are plotted in this virtual water flow network.(b) U.S. blue virtual groundwater exports ($CU_{Med}$) through all U.S. ports. Only flows > 0.1 Mm³ are plotted in this virtual water flow network. Mexico in addition to Africa and Eastern Asia are a notable destination for U.S. blue virtual groundwater exports through Gulf Coast ports. (c) U.S. blue virtual surface water exports ($CU_{Med}$) through all U.S. ports. Only flows > 0.1 Mm³ are plotted in this virtual water flow network. Eastern Asia is a notable destination for U.S. blue virtual surface exports through West Coast ports. Network maps are plotted with Gephi using the Map of Countries and GeoLayout plugins.**

76

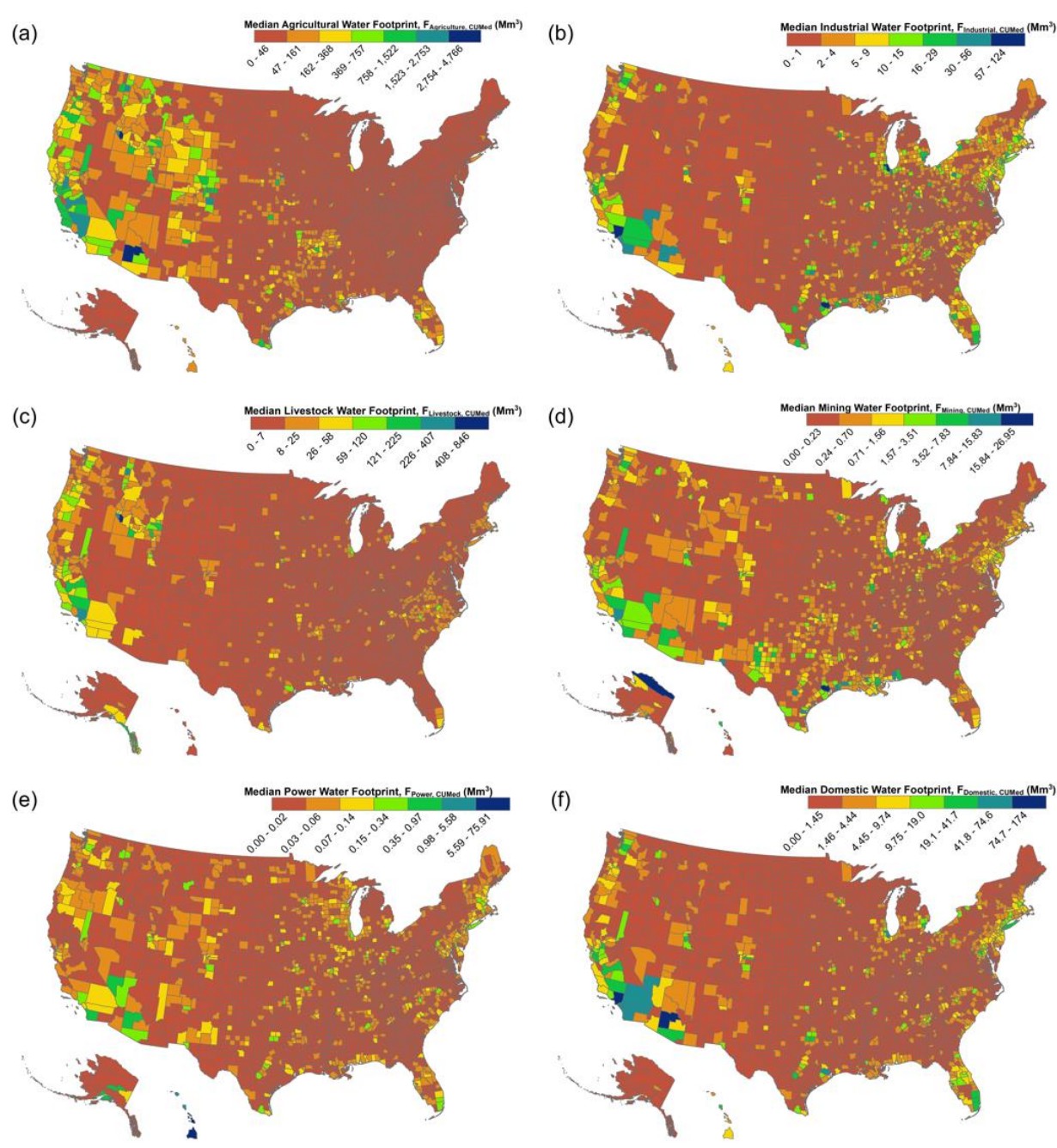

**Figure 6. (a) The county-level agricultural blue water footprint of the U.S. (b) The county-level industrial blue water footprint of the U.S. (c) The county-level livestock blue water footprint of the U.S. (d) The county-level mining blue water footprint of the U.S. (e) The county-level electrical power blue water footprint of the U.S. (f) The county-level domestic blue water footprint of the U.S.**

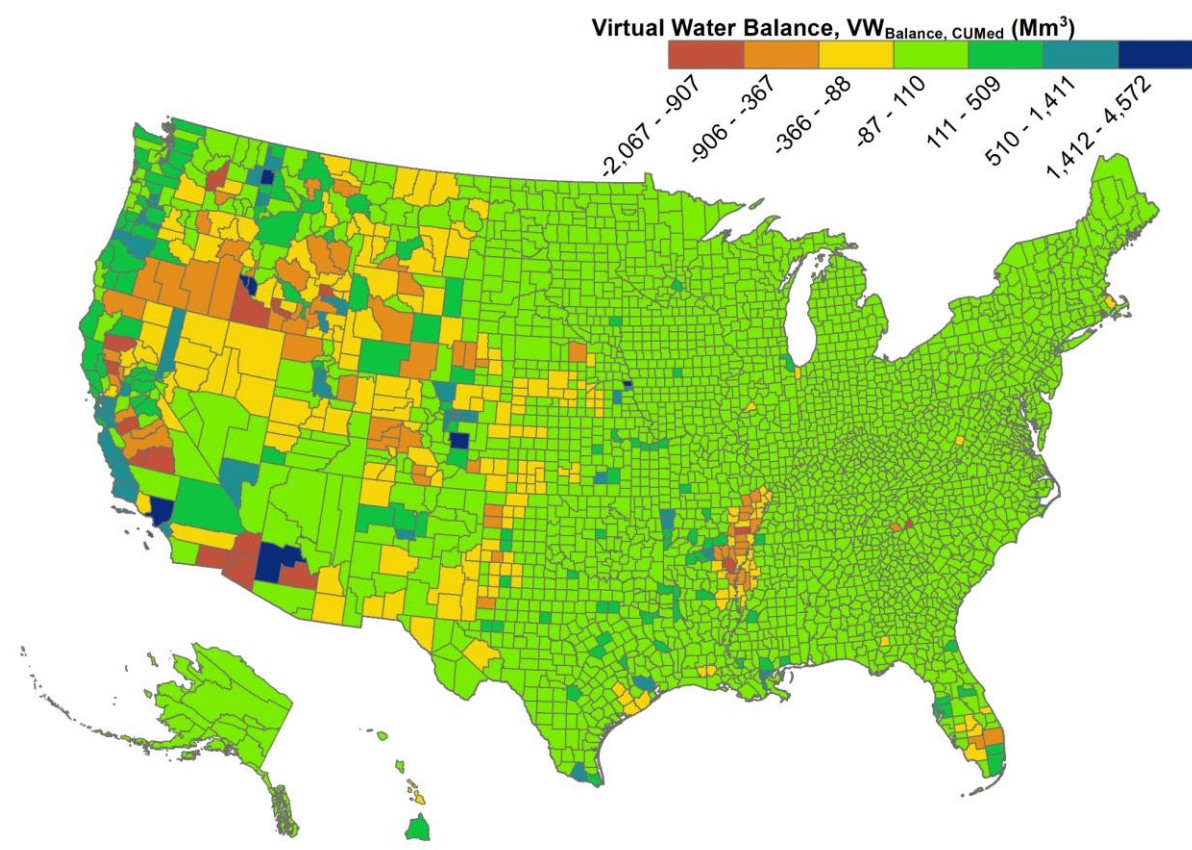

85

**Figure 7. The blue virtual water balance (VW$_{Balance, CUMed}$) for each U.S. county. Areas in the Southwest U.S., Central Valley of California, Snake River Valley, Mississippi Embayment, South Florida, South Texas, and the High Plains have virtual water outflows that outstrip virtual water inflows.**

90

91

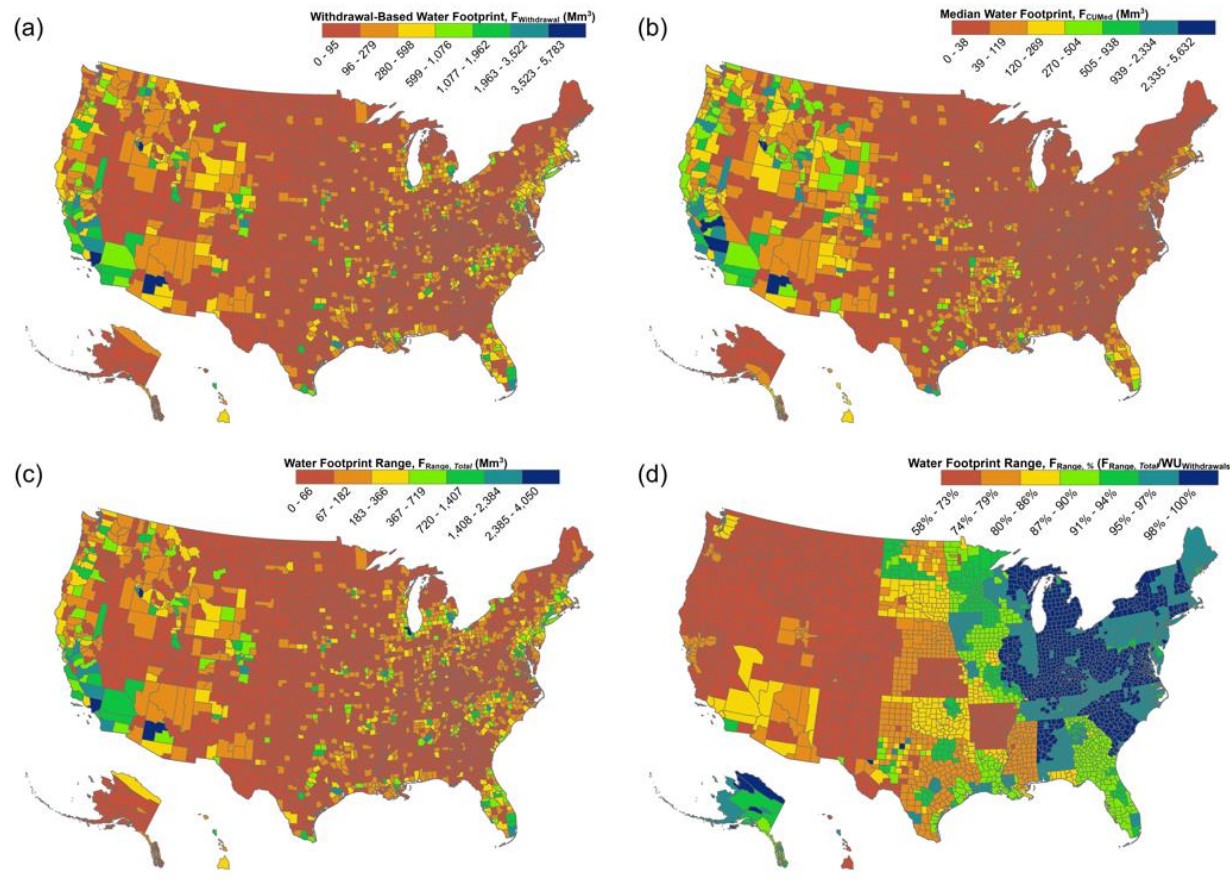

**Figure 8. (a) The annual withdrawal-based blue water footprint, $F_{Withdrawal}$ [Mm³], for U.S. Counties. (b) The annual med ($CU_{Med}$) blue water footprint, $F_{CUMed}$ [Mm³], for U.S. Counties. The minimum scenario was constructed applying minimum sector-level consumption coefficients. The range of uncertainty in the blue water footprint, $F_{Range}$ [Mm³], for U.S. Counties. $F_{Range}$ is computed as the range between the highest and lowest water footprints of the withdrawal-based and three consumption-based scenarios. Absolute water footprint uncertainties are highest in the west, but relative uncertainties are highest in the east. (d) Relative water footprint variation tends to increase in the Eastern United States and county-level water footprint uncertainty can range between 58.2 % in much of the Western United States to 99.9 % in parts of the Eastern United States.**

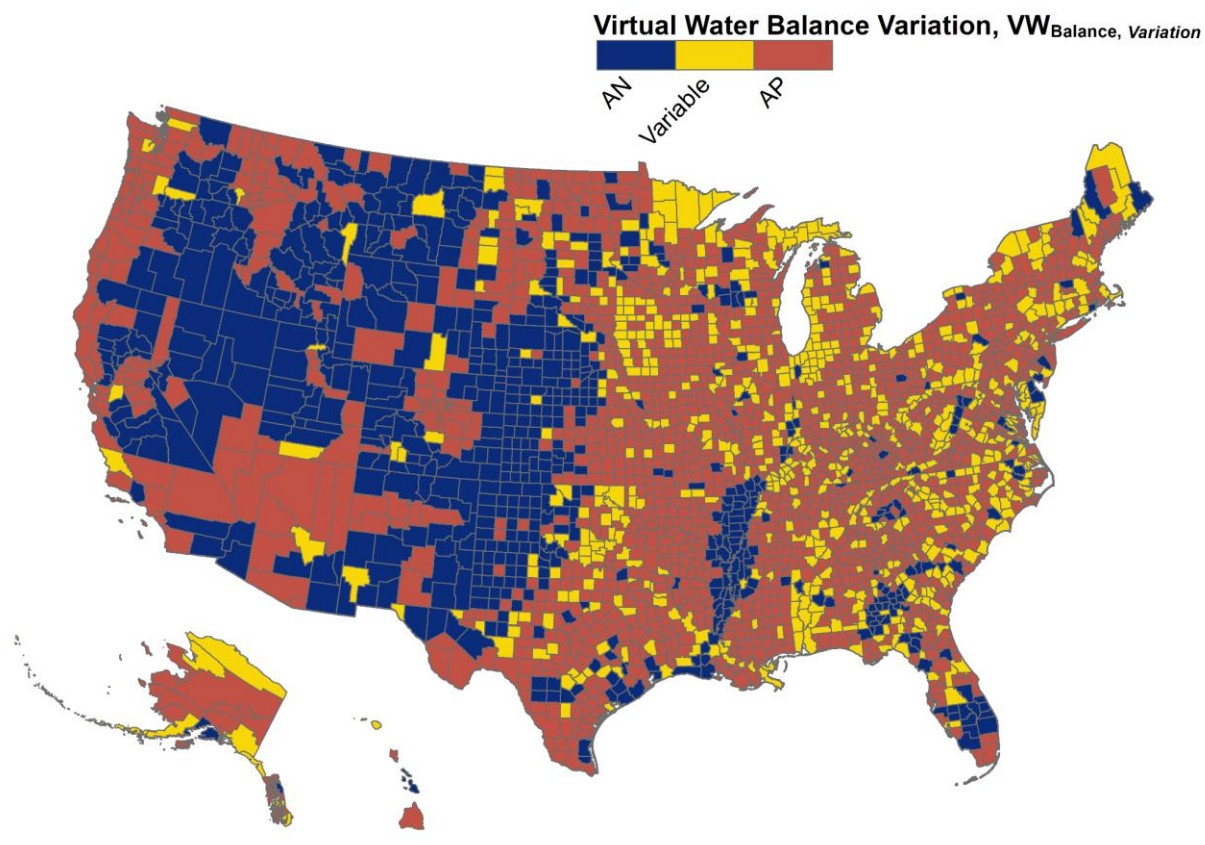

104

**Figure 9. For many counties, whether a county has a negative or positive virtual water**
**balance varies under the consumptive use scenarios. Counties in blue always have a**
**negative virtual water balance (AN) and virtual water outflows are always greater than**
**virtual water inflows. Counties in red always have positive virtual water balances (AP) and**
**virtual water inflows are always greater than virtual water outflows. Counties in yellow**
**have borderline-neutral net virtual water balances that depend on the consumptive use**
**uncertainty (Variable).**

112

113