# Peer review of "A Spatially Detailed Blue Water Footprint of the United States Economy"

_Hydrology and Earth System Sciences, 2017_

## Referee Comment (RC1) · Anonymous Referee #1 · 3 Jan 2018

General Comments: This article introduces a useful model to hydrologic sciences by attempting to capture the complex influence of economic activity on disparate water resources. The model contributes to the field of water footprinting and its public access is crucial to its policy relevance. With moderate revisions, I recommend this article be published in HESS.

Specific Comments (by line number in manuscript):

Title: I don't think you can call this model "economically complete" if you don't include the service sectors. Maybe something related to the "physical economy"?

10: in the abstract, it's unclear what "median" is the median of.

[Figure]

42: you haven't mentioned energy yet, so perhaps a sentence tying it into your hazards framing would be helpful.

57: what do you mean by "optimal" here? efficient? something more specific?

59: it seems appropriate to acknowledge here that non-managed green water still supports most of the world's food supply and virtual water trade. It's fine to focus your model only on blue water but this choice needs to be stated, justified, and situated within the larger picture of water-for-food. Somewhere in this paragraph could be a good place to do this.

66: before this paragraph it would be helpful to have a little primer and literature review on water footprinting. Referring to the water footprint method (i.e. Hoekstra et al), you are using both the "water footprint of production" and the "water footprint of consumption," so it would be good to mention these concepts. Also what is your/NWED's contribution to the literature? What's been done and how does your project fit in or push the boundaries of water footprinting?

71: this meta-statement could also mention some other aspects of the database, e.g. where it's housed, that it's publicly available, any other features.

81: the qualification after the comma seems unnessesary.

88: the inclusion of saline water is curious and could be clarified here so it's not conflated conceptually with your framing of freshwater hydro-economic vulnerability.

89: sentence missing "we leave"?

94: it should be mentioned that FAF is modeled data not surveyed data.

103: from what scale are FAZ linkages disaggregated?

112: please state that this attraction factor method assumes that all residents consume equally.

117: how are international water use factors derived?

121: seeing "water withdrawal" and "water footprint" in the same sentence raises questions, since WF is technically based on consumptive use. Please clarify here or state where you will clarify.

135: if you mention post-recession, you might also mention hydrological conditions, i.e. pre/post any notable droughts that could affect USGS withdrawal data.

156: number "of"

169: water footprints are not mentioned in this section, suggest deleting from section title.

172: it's unclear at this point why all of the attributes of FAF flows are important and how they're used. e.g. if you have tons, why do you need mode, $, and ton-miles?

190: outflows would not be summed, unless they're negative.

207: double "ofs".

247: I understand from earlier that commodity production can be simply, e.g. the number of livestock operations in a county, so how to you get from this to tons as the denominator of the intensity?

259: "tons of a c produced within a county according to disaggregated FAF data" is also (with comment at 249) confusing since FAF doesn't report production data.

306: I think you don't mean the "the" after the ";". Also, not sure what you mean here since EIA does report electricity consumption data.

369: this statistic could use a citation.

515: "circularly multiplied" could be "positive feedback loop"

542: "significant," not "signification"

[Figure]

588: specify that this is groundwater management in the US.

592: "to have"

654: after reading this section, and thinking about the magnitude of uncertainty in this model, I think you might want to revisit some of your claims in the results sections and conclusion. I realize this is very general, but just make sure you feel comfortable that what you say can be backed up by your model's findings.

733: "For example..." is not a complete sentence.

740: this is an interesting proposition, can you elaborate on what such governance systems might look like?

743: the mention of consumption coefficients seems to reference an earlier paragraph. Please clarify.

751: delete "and grey."

---

## Referee Comment (RC2) · Anonymous Referee #2 · 8 Feb 2018

This paper presents an update to a major data base (the National Water Economy Database v. 1.1, NWED), which the authors describe as a complete water footprint for the U.S., at least with respect to the nation's blue water flows. It merges several otherwise disparate sources of data, including those on water use, economic production and commodity trade, transportation, demographics, as well as on electricity generation and grid configuration. It analyzes the spatial configuration of the water footprint associated with economic activity, with a particular emphasis on urban-rural contrasts and the distribution of ports for international trade. It also evaluates sources of uncertainty in the estimation procedures and proposes next steps in future research on this topic.

Overall, this is a highly competent documentation of an important study. The authors make a convincing argument of the utility (in fact, necessity) of considering the hydro-economic redistribution of water in addition to its hydrologic dynamics, in order to accurately depict the present-day water resource picture. There is a more-or-less complete documentation of what the authors did, the assumptions they made, and the key results that they found. There was heavy lifting on the harmonization of the many diverse data sets they needed to incorporate into their analysis. The same was true of their aggregation/disaggregation approaches. While not 100% clear, they did lay out the logic for their particular simplifying and/or necessary assumptions.

I would view this work as at the current state-of-the-art and for this reason I would support its publication. At the same time, and in view of my otherwise supportive stand there are ways in which the manuscript stumbled and could be strengthened substantially.

While I have followed at least the general aspects of the virtual water debate for some time now, I found myself having to read and re-read sections of the text to make sure I understood exactly what the authors were trying to say. (Lines 490-92 giving the confusing contrast of urban vs rural water use via the terms "VW flows" vs "water footprints" is an example). A lexicon, cited at the beginning of the paper and placed at least into the supplement, showing clear definitions would have been extremely helpful. The authors could start with a definition of blue water, green water, grey water as a start to help the less initiated. Such definitions would certainly ensure a higher degree of readability among those not particularly well-versed in the VW literature.

Another major issue I had with the paper was the proliferation of maps, many of which were not particularly instructive. What was particularly striking was the absence of graphical elements that depicted the connectivity of trade and water flows, which was the heart of this paper. This detracts from its ultimate impact. For example, the maps in Figure 3, supposedly depicting the character of virtual water imports and exports fail to provide such connectivity. Perhaps showing and example of a major port and its
major inflows and outflows could solve this problem, repeating this for the other ports in the supplement. To/from arrows which are the mainstay of virtual water mapping (e.g. from Hoesktra, Oki, etc.) are missing. In addition, the many numerics contained in the tables could have been more efficiently presented as visuals (e.g., bar graphs or box and arrow diagrams).

In addition, it seems that the issue of non-renewable groundwater use would immediately come into play, but we see very little at all on this issue beyond some general statements. Given the vulnerability question addressed in Section 3.3., there needs to be some discussion of why overdraft was not addressed, or if not, how it would be addressed in a next stage of research. A perfect place to discuss this would be in the context of the dependence of international trade to certain countries on U.S. groundwater (lines 583-86).

The policy discussions were terse, not particularly well-thought out, or convincing (e.g., lines 646-52 and 668-72).

Additional comments:

- Some of the concepts presented bear a likeness to the ideas of Weiskel et al. that should be cited (see references below).

- The term "Economically Complete" in the title, unless it is some accepted nomenclature, is awkward at best.

- ABSTRACT: The term "mesoscale" needs to be changed or requires an additional modifier (i.e., county-level).

- INTRODUCTION: The stage-setting text dwelled far too much on the issue of drought and in the end never actually analyzed its impact.

- Section 2.2: Some discussion of the need to address sub-annual fluctuations in the relationship between water supply and use is warranted. That may best be addressed in the discussion on uncertainty.

- Section 2.7, end: A word on assuring mass conservation at the international scale is warranted.

- Line 559, 560: The total seems is comparable to Menkonnen & Hoekstra 2011 (Vol. 2 appendices) but who get a balance of -8800Mm3/y (from a greatly different set of gross values: ∼30,000Mm3/y in blue water imports and ∼39,000Mm3/y for exports). The VW import in Table 5 is an order of magnitude lower. Also, I was unable to reproduce the 6.3% figure for the "volume of domestic virtual water flow". Again, this may lead back to a nomenclature problem. Authors please explain and clarify.

- I found the discussion on how the ports in question are vulnerable (e.g., to hurricanes, earthquakes, etc.) a bit of a stretch in terms of links to water vulnerability. The links to water are simply matter-of-fact. One could argue that the listed disruptions, which after all are not water-related per se, are more important to the provision of global protein or computer chips or export $$ than to virtual water supplies.

- I found that the description of the urban-rural geography of water-economy links were a strength of this paper (e.g., identification of the importance of medium-sized U.S. cities to the hydro-economy; geography of the importance of different areas of the country to domestic vs int'l food provision (lines 565-76); discussion of insularity of some cities like Phoenix). But some of the writing on this raised a concern. On lines 498-501, there is the sentence: " Medium to small cities tend to be food processing hubs where farm goods are transformed into "food", and irrigated agricultural blue water footprints are registered in those small cities rather than in the large cities where the food is largely consumed." The authors need to comment on the "stranding" of the footprint accounting in the places where the food is "manufactured" rather than where it is consumed. What type of impact does this have on their overall conclusions?

EDITORIAL CHANGES:

Line 14: Change to "dominated by use at local and regional scales."

Line 75: Not known exactly what "composition" means.

Line 104: Are "attraction factors" the correct nomenclature?

Line 195: "taking"

Line 399: Missing reference

Lines 408-09. Sentence should read: "Total, surface water, and groundwater water footprints within a county match the standard Water Footprint Accounting definition of the water footprint of a geographic area (Hoekstra et al., 2012)."

Line 542: "significant"

Line 544: I'd wager that if the embodied water use by the livestock sector for feed were included this would not be so insignificant. Authors please comment.

Lines 548-50. The authors should comment on why these numbers are so small given the substantial withdrawals of fresh and saline water nationally by the thermoelectric sector.

Line 610: "is predominantly determined by the production, manufacture, and distribution of"

Lines 673-74: Cite Table SI 4-D?

REFERENCES:

Weiskel, P.K., Vogel, R.M., Steeves, P.A., DeSimone, L.A., Zarriello, P.J., and K.G. Ries, III, 2007, Water-use regimes: Characterizing direct human interaction with hydrologic systems, Water Resour. Res., April 2007. http://pubs.er.usgs.gov/publication/70030893

Weiskel, P.K., Brandt, S.L., DeSimone, L.A., Ostiguy, L.J., and Archfield, S.A., 2010, Indicators of streamflow alteration, habitat fragmentation, impervious cover, and water quality for Massachusetts stream basins: USGS SIR 2009 –5272, 70 p.,

http://pubs.usgs.gov/sir/2009/5272/

Weiskel, P. K., Wolock, D. M., Zarriello, P. J., Vogel, R. M., Levin, S. B., and Lent, R. M., 2014: Hydroclimatic regimes: a distributed water-balance framework for hydrologic assessment, classification, and management, Hydrol. Earth Syst. Sci., 18, 3855-3872, doi:10.5194/hess-18-3855-2014, 2014. http://www.hydrol-earth-syst-sci.net/18/3855/2014/hess-18-3855-2014.html

---

## Author Comment (AC1) · 9 Mar 2018

General Comments: This article introduces a useful model to hydrologic sciences by attempting to capture the complex influence of economic activity on disparate water resources. The model contributes to the field of water footprinting and its public access is crucial to its policy relevance. With moderate revisions, I recommend this article be published in HESS. Specific Comments (by line number in manuscript):

Title: I don't think you can call this model "economically complete" if you don't include the service sectors. Maybe something related to the "physical economy"?

> RESPONSE: Added the following sentence on Line 87 (pg. 5):
>
> "It is an economically complete water footprint database, to the extent possible, since it utilizes input water data that covers the majority of U.S. water withdrawal activities (Maupin et al., 2014). The service industry is included in NWED although we assume virtual water flows resulting from the service industries are *de minimus* compared to the commodity-producing sectors of the economy and thus do not estimate these flows (Rushforth and Ruddell, 2015)."
>
> Additionally, the title has been changed to: "**A Spatially Detailed Blue Water Footprint of the United States Economy.**"

10: in the abstract, it's unclear what "median" is the median of.

> RESPONSE: Line 10-12 (pg. 2). We've clarified "median" by adding the sentence: "Four virtual water accounting scenarios were developed with minimum (*Min*), median (*Med*), and maximum (*Max*) consumptive use scenarios and a withdrawal-based scenario."

42: you haven't mentioned energy yet, so perhaps a sentence tying it into your hazards framing would be helpful.

> RESPONSE: Line 38-41 (pg. 3): We added an appositive statement to specify that electricity and other types of energy are considered to be water-intensive goods: "Disruptions to the production and distribution of water-intensive goods, including electricity and other energy

sources, have the potential spread across seemingly disparate localities over short time periods and are inherently a coupled natural-human (CNH) system phenomenon (Liu et al., 2007)."

57: what do you mean by "optimal" here? Efficient? Something more specific?

RESPONSE: We deleted the word 'optimally'. Using the word optimally in this context is value-based and subjective. Starting at Line 51 (pg. 4) the new sentence reads: "Given the climatic, political, legal, geographical, and infrastructural constraints to developing new water supplies, which exist to varying extents worldwide, the potential solutions to systemic global water resources problems now lie in managing the scarcity, equity, and distribution of existing water resources through the global hydro-economic network rather than the large-scale development of new, physical sources of water (Gleick, 2003)."

59: it seems appropriate to acknowledge here that non-managed green water still supports most of the world's food supply and virtual water trade. It's fine to focus your model only on blue water but this choice needs to be stated, justified, and situated within the larger picture of water-for-food. Somewhere in this paragraph could be a good place to do this.

RESPONSE: To address this comment we added the following sentence at Line 55 (pg. 4): "Further, the importance of managing the scarcity, equity, and distribution of blue water resources only increases as rainwater becomes more variable because the majority of water used for food production in the U.S. is green water (rainwater) (Marston et al., 2018)."

Additionally, at Line 90 (pg. 5): "NWED, in its current form, is constrained to blue virtual water flows to focus on potential human-mediated intervention points in the U.S. hydro-economic network."

66: before this paragraph it would be helpful to have a little primer and literature review on water footprinting. Referring to the water footprint method (i.e. Hoekstra et al), you are using both the "water footprint of production" and the "water footprint of consumption," so it would be good to mention these concepts. Also what is your/NWED's contribution to the literature? What's been done and how does your project fit in or push the boundaries of water footprinting?

RESPONSE: Added a paragraph to the introduction starting at Line 66 (pg. 4):

"A water footprint is defined as the volume of surface water and groundwater consumed during the production of a good or service and is also called the virtual water content of a good or service (Mekonnen and Hoekstra, 2011a). Virtual water, also known as indirect water or embodied water, has been studied as a strategic resource for two decades as it allows geographic areas (country, state, province, city) to access more water than is physically available (Allan, 1998; Allan, 2003; Suweis et al., 2011; Dalin et al., 2012; Dang et al., 2015; Zhao et al., 2015; Marston et al., 2015). Using NWED data, water footprints of production and consumption can be calculated for U.S. counties, metropolitan areas, and states. A water footprint of production is the total volume of water consumed with a geographic boundary, including water consumption for local use less virtual water export (Mekonnen and Hoekstra, 2011b). A water footprint of consumption is water consumption for local use in addition virtual water import (Mekonnen and Hoekstra, 2011b).

This paper presents the first spatially-detailed and economically-complete blue water footprint database of a major country, the U.S., using data from the National Water Economy Database (NWED), version 1.1. The methodological innovations of NWED lie in trade flow downscaling through the novel data fusion of multiple U.S. Federal datasets. This process yields a complete, network-based water footprint database of surface water and groundwater with flexible geographic aggregation from the county-level to international-level for multiple transit modes and trade metrics. NWED is economically complete, to the extent possible, since it utilizes input water data that covers the vast majority of U.S. water withdrawal activities (Maupin et al., 2014). The service industry is included in NWED although we assume virtual water flows resulting from the service industries are *de minimus* compared to the commodity-producing sectors of the economy and thus do not estimate these flows (Rushforth and Ruddell, 2015). NWED contains four consumptive use scenarios – a withdrawal-based scenario, in addition to minimum, median, and maximum consumptive use scenarios. Currently, NWED is constrained to blue virtual water flows to focus on potential human-mediated intervention points in the U.S. hydro-economic network. This article is the publication of record for NWED, which is currently housed on the Hydroshare data repository (Rushforth and Ruddell, 2017)."

71: this meta-statement could also mention some other aspects of the database, e.g. where it's housed, that it's publicly available, any other features.

RESPONSE: See Line 92 (pg. 6): "This article is the publication of record for NWED, which is currently housed on the Hydroshare data repository (Rushforth and Ruddell, 2017)."

81: the qualification after the comma seems unnessesary.

RESPONSE: The qualification has been removed. Line 102 (pg. 6) now reads: "What is the current mesoscale uncertainty associated with blue water footprints in the United States given current data resources?"

88: the inclusion of saline water is curious and could be clarified here so it's not conflated conceptually with your framing of freshwater hydro-economic vulnerability.

RESPONSE: Included the following comment starting at Line 110 (pg. 6): "We include saline and reclaimed water to fully characterize the U.S. hydro-economy. Specifically, saline and reclaimed water is used as a direct substitute for freshwater use and is a significant percentage of saline water use for power generation in Florida and the largest nuclear power plant in the U.S., located in Arizona utilizes reclaimed water. Saline water is also becoming an important component of municipal water portfolios in California, Texas, Florida and other states. While the inclusion of saline and reclaimed water in NWED is not a doctrinaire interpretation of established blue water footprint methodologies, we do believe it is necessary to these water because these water types or no longer *de minimus* components of water supply. Additionally, if there are future constraints to utilizing saline or reclaimed water for power production, we will be able to anticipate the future added pressure on blue water resources. "

89: sentence missing "we leave"?

RESPONSE: Line 119 (pg. 7): "We leave green water footprints, and the aquatic ecosystem impacts of water use, to future work."

94: it should be mentioned that FAF is modeled data not surveyed data.

RESPONSE: Starting from Line 121 (pg. 6) the description of FAF data now reads as: "The hydro-economic network constructed in NWED is built from existing commodity flow networks and data, specifically the Freight Analysis Framework version 3.5 (FAF) developed by Oak Ridge National Laboratories for the U.S. Department of Transportation (Southworth et al., 2010H;([A-z])wang et al., 2016), which builds upon the U.S. Commodity Flow Survey by statistically modelling the flows of several out-of-scope commodity flows, notably farm-based agricultural flows, natural gas, crude petroleum, and waste."

103: from what scale are FAZ linkages disaggregated?

RESPONSE: Amended sentence starting at Line 130 (pg. 7) to read: "Details of the FAZs, how FAZ-level is derived, commodity classes, and transport modes have been documented elsewhere and, as such, will not be reproduced in this paper (Southworth et al., 2010; Hwang et al., 2016; U.S. Bureau of Transportation Statistics, 2017)."

112: please state that this attraction factor method assumes that all residents consume equally.

RESPONSE: Reworked the sentence starting at Line 163 (pg. 9) to read: "Currently, NWED uses population as the only attraction factor (U.S. Census Bureau, 2013), which is as a surrogate for county-level economic demand for commodities and that all residents consume goods equally. Population is an adequate attraction factor in the initial NWED version because it is a robust indicator available for every county in the U.S., but this attraction factor will be subject to further refinement as new NWED versions are developed."

117: how are international water use factors derived?

RESPONSE: This is addressed in the methods section 2.10. Clarification has been added to Line 147 (pg. 8): "Virtual water imports and exports were estimated using water intensity proxies and detailed in Section 2.10."

121: seeing "water withdrawal" and "water footprint" in the same sentence raises questions, since WF is technically based on consumptive use. Please clarify here or state where you will clarify.

RESPONSE: At Line 161 (pg. 9), added the sentence: "Four scenarios are developed from the USGS water input data: a withdrawal-based scenario (*Withdrawal*) and maximum (*Max*), median (*Med*), and minimum (*Min*) consumptive use scenarios." This sentence clarifies that we have developed virtual water flow data based on multiple consumptive use scenarios one which is withdrawal.

135: if you mention post-recession, you might also mention hydrological conditions, i.e. pre/post any notable droughts that could affect USGS withdrawal data.

Line 174 (pg. 9): Added the sentences: "Water withdrawal data for 2010 captures the beginning of Texas-North Mexico drought that lasted from 2010 to 2011 (Seager et al., 2014) and is situated between significant droughts in California between 2007 and 2009 (Christian-Smith et al., 2015) and 2011 to 2014 (Seager et al., 2015). It is possible that these two hydrologic

> droughts increased water groundwater withdrawals and consumption in the U.S. during 2010 calendar year in the southwestern and southcentral U.S."

156: number "of"

> Line 143 (pg. 8): "number of commodity-specific"

169: water footprints are not mentioned in this section, suggest deleting from section title.

> Line 225 (pg. 11): New Section Title, "NWED Naming Convention"

172: it's unclear at this point why all of the attributes of FAF flows are important and how they're used. e.g. if you have tons, why do you need mode, $, and ton-miles?

> RESPONSE: Added the following sentence at Line 243 (pg. 12): "NWED retains data for transport mode, tons, currency, and ton-miles as there are NWED use cases outside of virtual water accounting that may utilize mode-specific data or data on $ or *tm* flows."

190: outflows would not be summed, unless they're negative.

> RESPONSE: Clarified Line 248 (pg. 12) to read: "The water footprint of a geographic area ($F_{Total}$) is the sum of the direct water use ($WU$), virtual water inflows ($VW_{In}$), and virtual water outflows ($VW_{Out}$) (Hoekstra et al., 2012)."

> Due to the structure of NWED, VWIn or VWout may include 0-values and all values are stored as positive values and negatives are added by convention during the data processing.

207: double "ofs".

> RESPONSE: Fixed (pg. 3).

247: I understand from earlier that commodity production can be simply, e.g. the number of livestock operations in a county, so how to you get from this to tons as the denominator of the intensity?

> RESPONSE: The production factor is proxy to disaggregate activity within a sector. I clarified the livestock operations disaggregation statement in Line 140 (pg. 8) – that statement was inaccurate to the method used to produce version 1.1 (it was true for previous versions 0.9, 1.0). The, "production factor for the livestock sector is county-level livestock and animal sales for cattle, hogs, and poultry (USDA National Agricultural Statistics Service, 2012)." We feel this is adequate for the current version as cattle, hog, and poultry sales do not significantly overlap geographically. There is an underlying weight-basis to sales data, but average per head or animal weights were not utilized. This is under development for future versions of NWED.

259: "tons of a c produced within a county according to disaggregated FAF data" is also (with comment at 249) confusing since FAF doesn't report production data.

> RESPONSE: Reworded the Line 316 (pg. 16) to read: "The $VW_{In, Jm, c}$ that results from this process assigns water withdrawals to a commodity based on the tons of a *c* within a county according to the disaggregated FAF data." We removed the mention of produced. However, some commodity flows can be flow through, but farm-based flows as well as flows of oil and natural are tied to specific production activities.

306: I think you don't mean the "the" after the ";". Also, not sure what you mean here since EIA does report electricity consumption data.

> RESPONSE: Reworded the sentence starting at Line 390 (pg. 19) to read so that it is clearer that what is lacking in publically-available data is a model that shows coupled generation, transmission, and consumption at the mesoscale: "To harmonize with NWED and disaggregate ReEDS data from the balancing authority to the county-level, the model's production numbers are disaggregated proportionally using the heat content of fuel consumption for electricity for each county's power plants (Energy Information Administration, 2017) and electricity demand is disaggregated proportionally by population."

369: this statistic could use a citation.

> RESPONSE: Added a statistic and citation at Line 426 (pg. 20): "Saline and brackish water are non-trivial components of U.S. water use, comprising about 14% of total water withdrawals – specifically, power generation in Florida, mining in Texas and Oklahoma (Maupin et al., 2014)."

515: "circularly multiplied" could be "positive feedback loop"

> RESPONSE: Line 575 (pg. 27): circularly multiplied has been changed to positive feedback loop.

542: "significant," not "signification"

> RESPONSE: Line 602 (pg. 28): made correction.

588: specify that this is groundwater management in the US.

> RESPONSE: Clarified the sentence beginning at Line 664 (pg. 31): "Conversely, Canada, Latin America, Europe, and Asia and Oceania are more susceptible to surface water availability and drought but less susceptible to unsustainable groundwater management in the U.S."

592: "to have"

> RESPONSE: Corrected in Line 669 (pg. 31).

654: after reading this section, and thinking about the magnitude of uncertainty in this model, I think you might want to revisit some of your claims in the results sections and conclusion. I realize this is very general, but just make sure you feel comfortable that what you say can be backed up by your model's findings.

> RESPONSE: We understand this point, but at this moment we feel comfortable that our conclusions/discussion can be backed up by the model. However, we have reorganized the conclusions so that the discussion of uncertainty is the second paragraph (Line 781, page 36) and the following conclusions are therefore presented with uncertainty front and center so that any conclusions are tempered by the uncertainty discussion.
>
> Paragraph beginning at Line 781: "The uncertainty introduced by water use data and consumption coefficients demonstrate the great need for the development of region-specific, sector-level water use data and consumption coefficients for the entire U.S. For example, water footprint uncertainty is roughly 58 % to over 99 % of a county's total water footprint, which increases in the eastern United States. However, in absolute terms, consumption coefficient

variation is more important in the western U.S. due to the potentially large variation in virtual water outflows from the agricultural sector with largest blue water withdrawals. While we have presented results for the $CU_{Med}$ scenario in this paper, we must recognize the potentially large variation in water consumption that could exist compared to what is reported. Therefore, conclusions drawn from NWED data, as well as those drawn from the underlying water data, must recognize the large range of uncertainty with respect to water withdrawal and consumption in the U.S. Nevertheless, there are still general observable trends in U.S. virtual water flows and water footprints, which are presented below."

733: "For example..." is not a complete sentence.

RESPONSE: Fixed the sentence beginning with "For example" on line 837 (pg. 38).

740: this is an interesting proposition, can you elaborate on what such governance systems might look like?

RESPONSE: Expanded this thought with a sentence starting on Line 845 (pg. 39): "For example, downstream-driven, market-based supply chain governance of "soft" supply chains by major retailers and distributors; downstream-driven City-driven governance via their hard infrastructures (McManamay et al., 2017); upstream-driven, watershed- or river-driven governance wherein infrastructure managers consider how the services of their water propagate through the economy; or FEW governance where F, E, and W agents work together because these sectors have the largest footprints."

743: the mention of consumption coefficients seems to reference an earlier paragraph. Please clarify.

RESPONSE: On Line 854 (pg. 39): Added reference to section 3.5.

751: delete "and grey."

RESPONSE: Deleted grey from Line 751, but this paragraph has been moved into the end of the introduction to discuss the innovations presented in this paper.

**Response to Anonymous Reviewer #2:**

This paper presents an update to a major data base (the National Water Economy Database v. 1.1, NWED), which the authors describe as a complete water footprint for the U.S., at least with respect to the nation's blue water flows. It merges several otherwise disparate sources of data, including those on water use, economic production and commodity trade, transportation, demographics, as well as on electricity generation and grid configuration. It analyzes the spatial configuration of the water footprint associated with economic activity, with a particular emphasis on urban-rural contrasts and the distribution of ports for international trade. It also evaluates sources of uncertainty in the estimation procedures and proposes next steps in future research on this topic.

Overall, this is a highly competent documentation of an important study. The authors make a convincing argument of the utility (in fact, necessity) of considering the hydroeconomic redistribution of water in addition to its hydrologic dynamics, in order to accurately depict the present-day water resource picture. There is a more-or-less complete documentation of what the authors did, the assumptions they made, and the key results that they found. There was heavy lifting on the harmonization of the many diverse data sets they needed to incorporate into their analysis. The same was true of their aggregation/disaggregation approaches. While not 100% clear, they did lay out the logic for their particular simplifying and/or necessary assumptions.

I would view this work as at the current state-of-the-art and for this reason I would support its publication. At the same time, and in view of my otherwise supportive stand there are ways in which the manuscript stumbled and could be strengthened substantially.

While I have followed at least the general aspects of the virtual water debate for some time now, I found myself having to read and re-read sections of the text to make sure I understood exactly what the authors were trying to say. (Lines 490-92 giving the confusing contrast of urban vs rural water use via the terms "VW flows" vs "water footprints" is an example). A lexicon, cited at the beginning of the paper and placed at least into the supplement, showing clear definitions would have been extremely helpful. The authors could start with a definition of blue water, green water, grey water as a start to help the less initiated. Such definitions would certainly ensure a higher degree of readability among those not particularly well-versed in the VW literature.

> RESPONSE: A lexicon has been inserted as Appendix 2 on Line 898 (pg. 41):
>
> **Appendix 2: NWED Glossary**
>
> *Agricultural Sector*:  NWED sector comprised of farm-based activities to grow crops for food or industrial purposes. Irrigation is the primary water using activity in the agricultural sector (Maupin et al., 2014).
>
> *Attraction Factor*:  A fraction used to disaggregate commodity flows on the consumption side. In NWED 1.1, population is used as an attraction factor.  Each county within a FAZ is assigned a fraction equivalent to its percent of the total population.
>
> *County*:  A county or county equivalent (parish, borough, Washington D.C., or a independent city) is a sub-state geographic scale that is roughly equivalent to the mesoscale.
>
> *Destination*:  The geographic location where a commodity flow terminates.

[revised manuscript text omitted]

Another major issue I had with the paper was the proliferation of maps, many of which were not particularly instructive. What was particularly striking was the absence of graphical elements that depicted the connectivity of trade and water flows, which was the heart of this paper. This detracts from its ultimate impact. For example, the maps in Figure 3, supposedly depicting the character of virtual water imports and exports fail to provide such connectivity. Perhaps showing and example of a major port and its major inflows and outflows could solve this problem, repeating this for the other ports in the supplement. To/from arrows which are the mainstay of virtual water mapping (e.g. from Hoesktra, Oki, etc.) are missing. In addition, the many numerics contained in the tables could have been more efficiently presented as visuals (e.g., bar graphs or box and arrow diagrams).

RESPONSE: The choice to present choropleths was for simplicity.  Since there are >3140 counties in the U.S. trading multiple commodities (or even one), a map arrows connecting all of the flows, which also span orders of magnitude, would not convey any meaningful information. It would be a too busy to discern meaningful patterns.

However, we have added network maps of U.S. virtual water exports (Figs 4 and 5).  Figure 4 is accompanied with explanation in text starting at Line 656 (pg. 30). In-text reference to Fig. 5 was added at Line 658 (pg. 31).

"For example, virtual water flows through ports in the Houston metropolitan area are dominated by groundwater sources in the Ogallala Aquifer Region, the Mississippi Embayment aquifer system, and to a lesser extent the Central Valley of California, local groundwater sources, and southern Arizona (Fig. 4)."

[Figure]

**Figure 4. (a) U.S. blue virtual water exports ($CU_{Med}$) through ports in the Houston metropolitan area are sourced from the Central Valley of California, Central Utah and Northern Utah, Southern Arizona, the Ogallala Aquifer Region, South Texas and the Texas**

Gulf Coast, and the Mississippi Embayment aquifer region. Virtual water flows into the Houston ports and then is redistributed to the 8 world regions in NWED. Mexico is the largest recipient of virtual water flows from Houston ports. (b) Virtual groundwater flow through Houston ports is sourced from the Central Valley of California, Central Utah and Northern Utah, Southern Arizona, the Ogallala Aquifer Region, South Texas and the Texas Gulf Coast, and the Mississippi Embayment aquifer region. (c) Virtual surface water through Houston ports is sourced from the Central Valley of California, Southern California, the Phoenix Metropolitan Area, Northern Utah, and the Texas Gulf Coast. Network maps are plotted with Gephi using the Map of Countries and GeoLayout plugins.

[Figure]

**Figure 5. (a)** U.S. blue virtual water exports ($CU_{Med}$) through all U.S. ports. Only flows > 0.1 Mm³ are plotted in this virtual water flow network.**(b)** U.S. blue virtual groundwater exports ($CU_{Med}$) through all U.S. ports. Only flows > 0.1 Mm³ are plotted in this virtual water flow network. Mexico in addition to Africa and Eastern Asia are a notable destination for U.S. blue virtual groundwater exports through Gulf Coast ports. **(c)** U.S. blue virtual surface water exports ($CU_{Med}$) through all U.S. ports. Only flows > 0.1 Mm³ are plotted in this virtual water flow network. Eastern Asia is a notable destination for U.S. blue virtual surface exports through West Coast ports. Network maps are plotted with Gephi using the Map of Countries and GeoLayout plugins.

In addition, it seems that the issue of non-renewable groundwater use would immediately come into play, but we see very little at all on this issue beyond some general statements. Given the vulnerability question addressed in Section 3.3., there needs to be some discussion of why overdraft was not addressed, or if not, how it would be addressed in a next stage of research. A perfect place to discuss this would be in the context of the dependence of international trade to certain countries on U.S. groundwater (lines 583-86). The policy discussions were terse, not particularly well-thought out, or convincing (e.g., lines 646-52 and 668-72).

> RESPONSE: Added a sentence on groundwater overdraft in Line 661 (pg. 31): "While we do not address surface or ground water sustainability, vulnerability, or overdraft specifically in this paper, it is certainly desirable to combine these results with quantification of water storage and water availability, for the purpose of policy analysis."

> With regard to the policy discussions, the focus of this paper is not so much policy, but documenting NWED and its features, and the straightforward application of NWED to water footprint results. Future research will focus on policy evaluations, which are out of scope for this paper.

> Added section 3.7 on Line 761 (pg. 35), "Temporal Uncertainty":

> "As mentioned previously, the NWED data are limited in representativeness to roughly the 2010 – 2012 post-recession timeframe but are not precisely linked to a single year. Temporal uncertainty is introduced by utilizing annual timescale data. Given this, NWED data are more directly relevant to surface water management than to groundwater management because surface water has months to a few years of storage, and groundwater has centuries of storage, but in the future we could use this data to analyze sustainability and vulnerability of water usage."

Additional comments:

- Some of the concepts presented bear a likeness to the ideas of Weiskel et al. that should be cited (see references below).

> RESPONSE: This review has brought the Weiskel et al. work to our attention and I think there is a lot of potential for collaboration between the NWED data and the approaches that Weiskel at al. have taken with hydroclimatic modelling and classification. We've added citations to the Weiskel works on Line 186 (pg. 10) – "However, annual water withdrawal and consumption data could be disaggregated to the month scale using median monthly demand curves (Archfield et al., 2009; Weiskel et al., 2010)." – and Lines 815, and 818 (pg. 37-38) – "Further work on characterizing county-level virtual water flows can extend the logic developed by frameworks to characterize catchment-level water use regimes (Weiskel et al., 2007) to hydro-economic networks. Specifically, NWED data can provide a socio-hydrological extension to previous work on hydroclimatic regime classification in the U.S. (Weiskel et al., 2014)."

- The term "Economically Complete" in the title, unless it is some accepted nomenclature, is awkward at best.

RESPONSE: Added the following sentence on Line 87 (pg. 5):

"It is an economically complete water footprint database, to the extent possible, since it utilizes input water data that covers the majority of U.S. water withdrawal activities (Maupin et al., 2014). The service industry is included in NWED although we assume virtual water flows resulting from the service industries are *de minimus* compared to the commodity-producing sectors of the economy and thus do not estimate these flows (Rushforth and Ruddell, 2015)."

Additionally the title has been changed to "**A Spatially Detailed Blue Water Footprint of the United States Economy**"

ABSTRACT: The term "mesoscale" needs to be changed or requires an additional modifier (i.e., county-level).

RESPONSE: Qualifer added in Line 4 (pg. 2): "NWED utilizes multiple mesoscale (county-level) federal data resources…"

- INTRODUCTION: The stage-setting text dwelled far too much on the issue of drought and in the end never actually analyzed its impact.

RESPONSE: The goal of this paper is to introduce a methodology and dataset that is a step forward in this direction. Future papers and datasets will evaluate specific aspects of drought and policy. See previous comment with respect to groundwater overdraft. Therefore, we've reduced the focus on drought in the introduction and reworked the introduction to focus on NWED, what it contains, and its innovations. This addition to the introduction can be found on Line 66 (pg. 4).

- Section 2.2: Some discussion of the need to address sub-annual fluctuations in the relationship between water supply and use is warranted. That may best be addressed in the discussion on uncertainty.

RESPONSE: Added the following paragraph to Section 2.2 (Line 182, pg. 9):

"The current version of NWED has an annual resolution due to a lack of comprehensive, sub-annual county-level data. While economic data are available at sub-annual timescales, often quarterly, water withdrawal data are not. However, annual water withdrawal and consumption data could be disaggregated to the month scale using median monthly demand curves (Archfield et al., 2009; Weiskel et al., 2010). This lack of data availability does present challenges because there are substantial sub-annual fluctuations in water withdrawal and consumption. Water demands for agriculture and power are highly seasonal and neither the beginning nor the end of a drought coincides with calendar years. For example, the Texas-North Mexico drought began in the latter half of 2010 (Seager et al., 2014). As we further develop NWED, we will develop methods to address this shortcoming, but for now are limited to the annual timescale."

- Section 2.7, end: A word on assuring mass conservation at the international scale is warranted.

RESPONSE: See the paragraph starting at Line 343 (pg. 17): "After this step, there is a final mass balance check to ensure NWED freight totals match underlying FAF data and water data match

underlying USGS data. NWED contains data detailing 3,142 counties trading 43 commodities with 3,142 counties, as well as 8 world regions, over 8 transport modes and each commodity trade linkage is measured by 15 metrics (The full list of metrics is in the Appendix, A2)."

- Line 559, 560: The total seems is comparable to Menkonnen & Hoekstra 2011 (Vol. 2 appendices) but who get a balance of -8800Mm3/y (from a greatly different set of gross values: ~30,000Mm3/y in blue water imports and ~39,000Mm3/y for exports). The VW import in Table 5 is an order of magnitude lower. Also, I was unable to reproduce the 6.3% figure for the "volume of domestic virtual water flow". Again, this may lead back to a nomenclature problem. Authors please explain and clarify.

Added a new section, section 3.6, which starts on Line 739 (pg. 34) entitled, "Uncertainty in International Virtual Water Flow" which reads:

"As mentioned in Section 3.4, there are several potential reasons for the discrepancy in the magnitude of virtual water flows.  First, there are differences in the underlying source data for international trade and water use. NWED utilizes commodity flows modeled by FAF, which itself utilizes Census Foreign Trade Data for 2010 (Southworth et al., 2010; Hwang et al., 2016),  while benchmark international virtual water trade studies utilized trade data from the International Trade Centre averaged between 1996-2005 (Water Footprint Network, 2013). Additionally, the source water data for the U.S. are different. NWED utilizes USGS water withdrawal data, which is self-reported with state-level variations (Marston et al., 2018; Maupin et al., 2014), benchmark international virtual water trade studies utilized CROPWAT modeling (Water Footprint Network, 2013). Secondly, despite controlling for port influences, it is likely that more virtual water is attributed to ports than necessary, which would dampen international virtual water flows in NWED. NWED has difficulty handling 'flow through' virtual waters flow that would be otherwise assigned to a point of final consumption. In this case, a flow through entity may be assigned virtual water flow at the port or another distribution hub.  Lastly, previous international virtual water studies included the water use of inputs in the virtual water flow of a commodity, e.g., the water consumption for animal feed as part of animal products related virtual water flow. A method to handle this is under development for the next version of NWED. While there are disadvantages to the current method in which international trade is modeled in NWED, methods to improve this aspect of the data product are ongoing and there is data structure in place to merge additional international trade flow datasets with the current NWED data structure.

- I found the discussion on how the ports in question are vulnerable (e.g., to hurricanes, earthquakes, etc.) a bit of a stretch in terms of links to water vulnerability. The links to water are simply matter-of-fact. One could argue that the listed disruptions, which after all are not water-related per se, are more important to the provision of global protein or computer chips or export $$ than to virtual water supplies.

RESPONSE: I agree with the point that port disruption are "are not water-related per se, are more important to the provision of global protein or computer chips or export $$ than to virtual water supplies." However, U.S. ports are not homogenous and different ports access different parts of the U.S. for different goods that utilize different types of water sources. A disruption to a specific port may pose create vulnerabilities as redirecting the flow of goods may either be too

costly or take an extended period of time. However, we understand the point of the commenter.

We've deleted the last two paragraphs of section 3.3, so there is no mention of port vulnerability. These paragraphs are tangential to the scope of the paper.

- I found that the description of the urban-rural geography of water-economy links were a strength of this paper (e.g., identification of the importance of medium-sized U.S. cities to the hydro-economy; geography of the importance of different areas of the country to domestic vs int'l food provision (lines 565-76); discussion of insularity of some cities like Phoenix). But some of the writing on this raised a concern. On lines 498-501, there is the sentence: "Medium to small cities tend to be food processing hubs where farm goods are transformed into "food", and irrigated agricultural blue water footprints are registered in those small cities rather than in the large cities where the food is largely consumed." The authors need to comment on the "stranding" of the footprint accounting in the places where the food is "manufactured" rather than where it is consumed. What type of impact does this have on their overall conclusions?

RESPONSE: The reviewer brings up a valid point regarding medium cities. However, we do not think our conclusions would change as the conclusions were drawn from the results presented. NWED is a versioned dataset, so development of better methods to address specific problem are always under development. The issue of assigning virtual water to where food is manufacturing instead of where it is consumed is one we are working on currently under development. That being said, the areas where food is processed or packaged (typically medium to small cities) are vulnerable to hydro-economic disruption directly through employment and food, so it is worthwhile to evaluate the system at this point in the supply chain in addition to farm-to-table.

To address this comment we've added elaborated on the line of text the commenter points on Line 558 (pg. 26): "Medium to small cities tend to be food processing hubs where farm goods are transformed into 'food.' and NWED assigns irrigated agricultural blue water footprints to these hubs. We recognize that this framing of the economy emphasizes different parts of the supply chain than previous studies and are developing methods for supply chain harmonization."

EDITORIAL CHANGES:

Line 14: Change to "dominated by use at local and regional scales."

RESPONSE: Updated sentence starting on Line 16 (pg. 2).

Line 75: Not known exactly what "composition" means.

RESPONSE: Updated question in Line 97 (pg. 6) to read: "How does the degree to which a geographic area is urban or rural affect water footprints, virtual water flows, and net hydro-economic dependencies?"

Line 104: Are "attraction factors" the correct nomenclature?

> RESPONSE:  The term attraction factor is utilized by source data documentation.

Line 195: "taking"

> RESPONSE: Remove "taking" from Line 253 (pg. 13).

Line 399: Missing reference

> RESPONSE: Added reference to reference list Line 1123 (pg. 47).

Lines 408-09. Sentence should read: "Total, surface water, and groundwater water footprints within a county match the standard Water Footprint Accounting definition of the water footprint of a geographic area (Hoekstra et al., 2012)."

> RESPONSE: Updated sentence starting on Line 468 (pg. 22): "Total, surface water, and groundwater water footprints within a county match the standard Water Footprint Accounting definition of the water footprint of a geographic area (Hoekstra et al., 2012).

Line 542: "significant"

> RESPONSE: Line 602 (pg. 28): made correction.

Line 544: I'd wager that if the embodied water use by the livestock sector for feed were included this would not be so insignificant. Authors please comment.

> RESPONSE: Included the comment starting on Line 589 (pg. 28): "Due to the structure of the underlying commodity flow dataset, the livestock sector only includes on-site water consumption at livestock operations. Inclusion of water usage for livestock feed would, no doubt, increase virtual water transfers related to the livestock sector and a method to do so is under development for the next NWED version."

Lines 548-50. The authors should comment on why these numbers are so small given the substantial withdrawals of fresh and saline water nationally by the thermoelectric sector.

> RESPONSE: While there are large water withdrawals associated with the power sector, water consumption is relatively low compared to other sectors. Since the results presented are consumption-based, they are small relative to the other sectors.

> Sentence added on Line 614 (pg. 29): "While there are large water withdrawals associated with the power sector, water consumption is relatively low compared to other sectors. Since the results presented are for the $CU_{Med}$ scenario, the power sector virtual water flows are small relative to the other sectors."

Line 610: "is predominantly determined by the production, manufacture, and distribution of"

> RESPONSE: Updated sentence on Line 673 (pg. 31) to read: "The U.S. water footprint is predominantly determined by the production, manufacture, and distribution of food."

Lines 673-74: Cite Table SI 4-D?

RESPONSE: Added the reference to the table on Line 737 (pg. 34).

[revised manuscript text omitted]

---

## Author Comment (AC3) · 9 Mar 2018

In the attached supplement, please find the revised manuscript for hess-2017-650, "A Spatially Detailed and Economically Complete Blue Water Footprint of the United States." All changes from both RC1 and RC2 have been tracked.

Please also note the supplement to this comment: https://www.hydrol-earth-syst-sci-discuss.net/hess-2017-650/hess-2017-650-AC3-supplement.pdf

650, 2017.